

# Climate and stratospheric ozone during the mid-Holocene and Last Interglacial simulated by MRI-ESM2.0

Yasuto Watanabe[1], Makoto Deushi[1], Kohei Yoshida[1]

[1]Meteorological Research Institute, Japan Meteorological Agency, Tsukuba, 305-0052, Japan

*Correspondence to*: Yasuto Watanabe (ywatanabe@mri-jma.go.jp)

**Abstract.** The climates of the mid-Holocene (MH) and Last Interglacial (LIG) are characterised by warm periods caused by astronomical forcing and climate feedback. One potential feedback is variation in the stratospheric ozone, the influence of which would extend down to the troposphere, potentially affecting the climate. However, understanding the role of changes in the stratospheric ozone during past warm interglacial periods is limited to MH conditions. Here, we employ MRI-ESM2.0,

an Earth system model with an iterative ozone model, and simulate the climate and atmospheric ozone during the MH and LIG. We show that the vertical and seasonal changes of stratospheric ozone in the LIG exhibited a stronger variation in the stratospheric ozone near the South Pole compared to that in the MH, indicating that both obliquity and precession forcings affect the stratospheric ozone distributions. We further show that its impact on the zonal mean surface air temperature is small, while it may affect surface air temperature regionally. These results advance the understanding of the dynamics of

atmospheric ozone to astronomical forcing during the warm interglacials.

## 1 Introduction

Investigation of past interglacials is expected to provide fundamental knowledge of feedback mechanisms that interact between the atmosphere, sea ice, and land surface under warmer climatic conditions (Otto-Bliesner et al. 2021). The climates of the mid-Holocene (MH; 6 ka) and the Last Interglacial (LIG; 127 ka) are characterised by warm and humid land climate

conditions, which would be attributed primarily to the different configurations of astronomical forcing (i.e., obliquity, precession, and eccentricity) that results in latitudinal and seasonal changes in insolation (Figure 1) (Braconnot et al. 2007, 2012; Otto-Bliesner et al. 2017, 2021). Paleoclimate proxy data indicate that the global mean surface air temperature was ~0.7 and ~1.3 K higher than the preindustrial (PI) conditions in the MH and LIG, respectively (Turney and Jones 2010; Marcott et al. 2013; Fischer et al. 2018; Kaufman et al. 2020b, a; Kaufman and Broadman 2023). The warming is especially

enhanced over land areas, which implies the amplification by feedback mechanisms between the atmosphere and sea ice over the Arctic Ocean and/or climate and vegetation feedback over land areas of the Northern Hemisphere high latitudes (Otto-Bliesner et al. 2006; O'ishi et al. 2021). This would have also contributed to the warming over Greenland and Antarctica, causing a sea level rise of 6–9 m during the LIG (Jouzel et al. 2007; NEEM Community Members 2013; Dutton et al. 2015). However, many climate models do not simulate the higher global annual mean surface air temperature inferred from the



paleoclimate proxy, which is called the Holocene temperature conundrum (Masson-Delmotte et al. 2013; Liu et al. 2014; Brierley et al. 2020; Kaufman and Broadman 2023) and its cause has been vigorously debated (Liu et al. 2014, 2018; Hopcroft and Valdes 2019; Park et al. 2019; Bova et al. 2021; Zhang and Chen 2021; Thompson et al. 2022; Laepple et al. 2022; Kaufman and Broadman 2023). Therefore, high latitude processes over land and ocean are critical for further understanding the climate warming during the past interglacials.

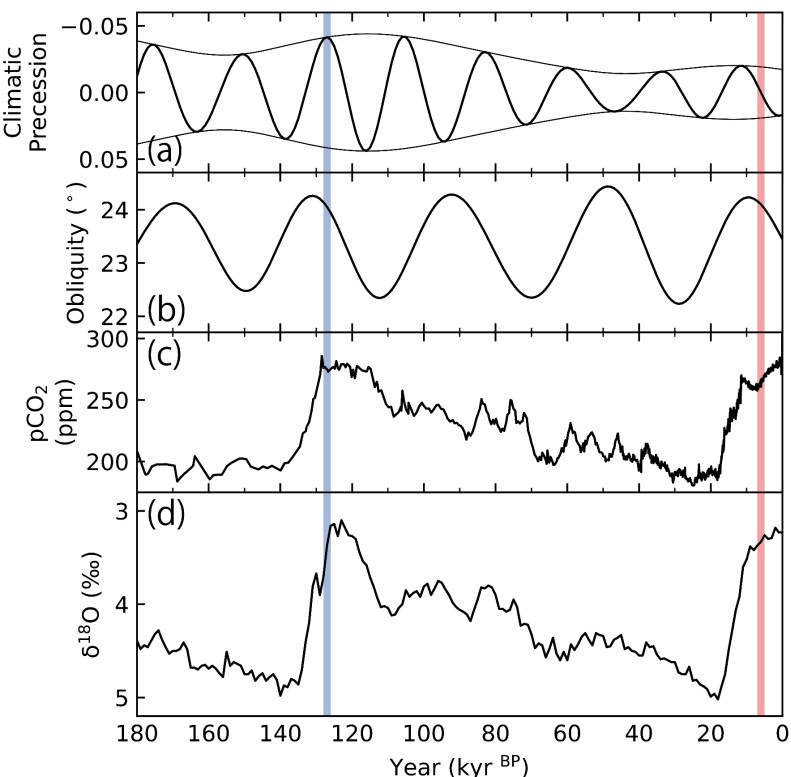


**Figure 1: Figure 1. Changes in (a) eccentricity (envelope curve) and climatic precession (Berger and Loutre 1991), (b) obliquity of the Earth (Berger and Loutre 1991), (c) atmospheric $p$CO$_2$ (Bereiter et al. 2015), and (d) stacked oxygen isotope signatures (Lisiecki and Raymo 2005) for the last 180 kyr.**

40        One of the potential factors that can affect the climate over high latitudes is changes in the stratospheric ozone (Thompson and Wallace 2000; Noda et al. 2017). It has been shown that changes in the ozone hole around the South Pole (SP) during the late 20th century affect the climate around Antarctica by changing the atmospheric structure and circulation patterns over those regions. Specifically, the depletion of the stratospheric ozone around the SP cools the stratosphere and strengthens the westerly jet and the positive Southern Annular Mode (Thompson and Wallace 2000; Noda et al. 2017). This





has been pointed out to further affect the distribution of sea ice around Antarctica because the changes in jet strength affect the Ekman transport in the Southern Ocean (SO) and the warming of the surface ocean (Sigmond and Fyfe 2010, 2014; Bitz and Polvani 2012; Smith et al. 2012; Noda et al. 2017).

A previous study employed the Earth system model MRI-ESM1, which couples an interactive ozone model, to estimate the response of the ozone layer during the MH (Noda et al. 2017). This study showed that the positive ozone

anomaly in the upper stratosphere around the South Pole in the austral summer during the MH caused by the different astronomical forcing would propagate to the lower stratosphere in the austral winter. This would increase the air temperature and weaken the southern westerly jet, contributing to the retreat of sea ice in the Southern Ocean during the MH. Thus, consideration of the interactive ozone layer in climate models would potentially affect the estimated climate state near the southern polar region. Specifically, the effect of ozone distribution changes may strengthen further and enhance the climate

change during the LIG owing to the effect of precession forcing under the high eccentricity during this period (Fig. 1), which may have contributed to the warming around Antarctica during the LIG. In the latest *Paleoclimate Modelling Intercomparison Project* (PMIP4), however, only one model, *Meteorological Research Institute Earth System Model 2.0* (MRI-ESM2.0) (Yukimoto et al. 2019), couples an interactive ozone chemistry model. While several studies have investigated the role of changes in ozone distribution during the Last Glacial Maximum (Rind et al. 2009; Murray et al.

2014; Noda et al. 2018; Wang et al. 2020, 2022), the validity of the role of ozone layer changes in response to different astronomical forcings during the MH pointed out by Noda et al. (2017) and its impact on the LIG climate system remains ambiguous.

In this study, we employed MRI-ESM2.0 and simulated the climate and the atmospheric ozone distribution under the MH and LIG conditions. Here, we discuss the atmospheric structure and circulation pattern with a specific focus on high

latitude regions in the Southern Hemisphere and the ozone distribution changes during the MH and LIG and their impact on the climate and atmospheric circulation.

## 2 Method

In this study, we employed MRI-ESM2.0 (Yukimoto et al. 2019), which couples the atmospheric general circulation model (AGCM) with land processes MRI-AGCM3.5, the aerosol model MASINGAR mk-2r4c (Tanaka et al. 2003), the

atmospheric chemistry climate model MRI-CCM2.1 (Deushi and Shibata 2011), and the ocean and sea ice model MRI.COMv4. The horizontal resolutions of the model grid are TL159 (~120 km) for MRI-AGCM3.5, TL95 (~180 km) for MASINGAR mk-2r4c, and T42 (~280 km) for MRI-CCM 2.1 (Yukimoto et al. 2019). The number of vertical layers of the AGCM, aerosol, and ozone models is 80. The MH conditions were previously calculated for PMIP4, but the LIG conditions have not been calculated.

We conducted the simulation using MRI-ESM2.0 under the PI conditions (Eyring et al. 2016; Menary et al. 2018) for 201 model years starting from the reference state for the PI conditions submitted to CMIP6 to achieve a new steady state



reflecting the minor improvement of the model (*PIcontrol*) (Table 1). We also conducted simulations under the MH and LIG conditions for 201 model years (*MHcontrol* and *LIGcontrol*, respectively) (Table 1), starting from the calculation under MH conditions submitted to PMIP4. In these calculations, the last 150 years were used for the analysis and the first 51 years were

omitted as a spin-up period. These calculations were conducted following the PMIP4 protocol (Otto-Bliesner et al. 2017). To elucidate the effect of the changes in the ozone distributions in the atmosphere, we also ran the model under the PI, MH, and LIG conditions using the monthly ozone climatology of the last 150 years of the calculation for the PI conditions (*PInochem*, *MHnochem*, and *LIGnochem*, respectively) (Table 1). These calculations were conducted for 201 years, and the last 150 years were analysed. The monthly outputs were corrected to consider the changes in the length of months (Bartlein and

Shafer 2018).

**Table 1: List of numerical experiments conducted in this study.**

| Name | Astronomical forcing | Greenhouse gases | Duration | Atmospheric ozone distributions |
|---|---|---|---|---|
| *PIcontrol* | 1,850 CE | $CO_2$: 284.3 ppm <br> $CH_4$: 808.2 ppb <br> $NO_2$: 273.0 ppb | 51 years (spinup) + 150 years | Dynamic |
| *MHcontrol* | 6 ka | $CO_2$: 264.4 ppm <br> $CH_4$: 597 ppb <br> $NO_2$: 262 ppb | 51 years (spinup) + 150 years | Dynamic |
| *LIGcontrol* | 127 ka | $CO_2$: 275 ppm <br> $CH_4$: 685 ppb <br> $NO_2$: 255 ppb | 51 years (spinup) + 150 years | Dynamic |
| *PInoChem* | 1,850 CE | $CO_2$: 284.3 ppm <br> $CH_4$: 808.2 ppb <br> $NO_2$: 273.0 ppb | 51 years (spinup) + 150 years | Monthly climatology of the last 150 years of the *PIcontrol* experiment |
| *MHnoChem* | 6 ka | $CO_2$: 264.4 ppm <br> $CH_4$: 597 ppb <br> $NO_2$: 262 ppb | 51 years (spinup) + 150 years | Monthly climatology of the last 150 years of the *PIcontrol* experiment |
| *LIGnoChem* | 127 ka | $CO_2$: 275 ppm <br> $CH_4$: 685 ppb <br> $NO_2$: 255 ppb | 51 years (spinup) + 150 years | Monthly climatology of the last 150 years of the *PIcontrol* experiment |




## 3 Results

### 3.1 Climate states under the MH and LIG conditions

The 150-year mean value of the surface (2 m) air temperature and total precipitation rate during the MH and LIG are shown
in Fig. 2. The global annual mean surface air temperatures were ~13.7 and ~13.8 ˚C under the MH and LIG conditions,
respectively, and both were lower than that of our PI-control experiment (~14.1 ˚C). Although the paleoclimate proxy
indicates a warmer environment during the MH and LIG, the model does not simulate the higher global annual mean surface

air temperature, which is similar to the results from other climate models (Liu et al. 2014; Brierley et al. 2020). The decrease
in the global annual mean surface air temperature is attributed primarily to the decrease in the temperature in North Africa
and India owing to the northern shift of the Asia–Africa monsoon system caused by the different sets of orbital parameters,
as seen in many climate models (Brierley et al. 2020; Otto-Bliesner et al. 2021). This is also reflected in the increased total
precipitation in these regions in both the MH and LIG (Fig. 2b and 2d).

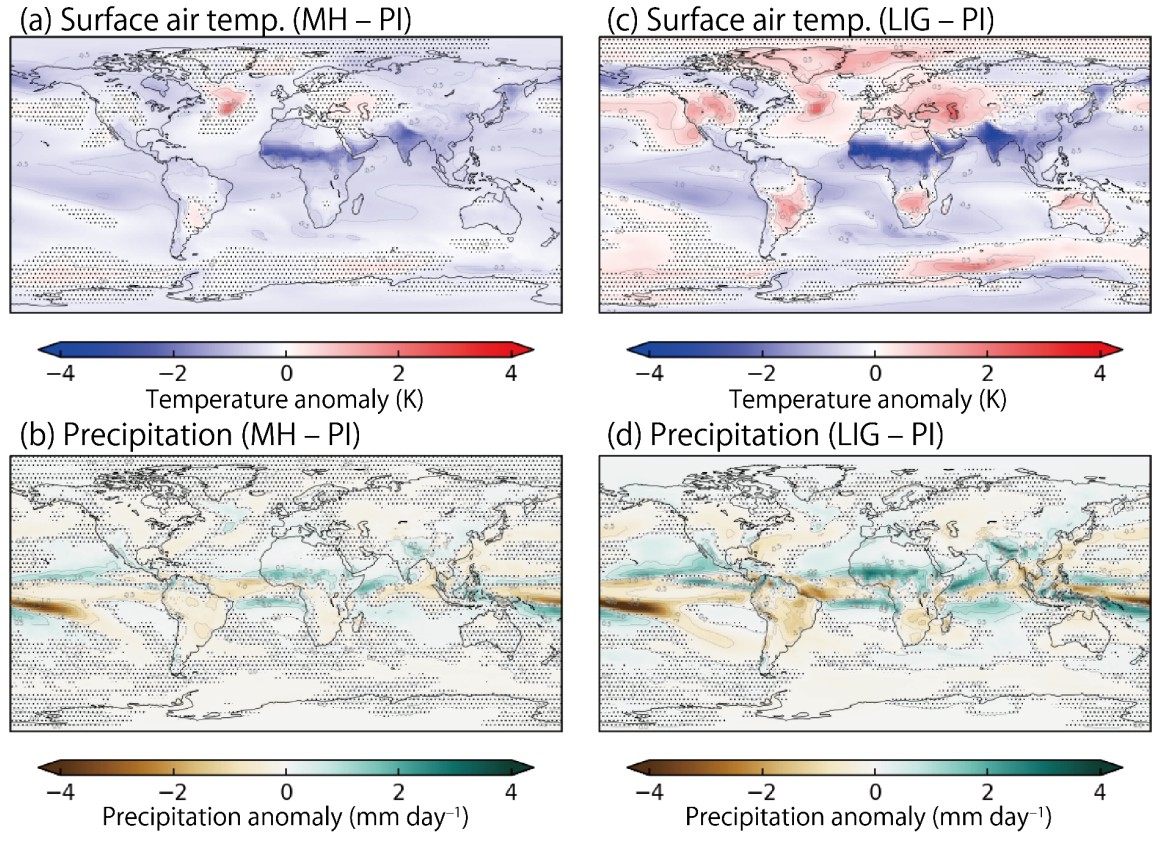


**Figure 2: Surface (2m) air temperature anomaly (a, c) and total precipitation anomaly (b, d) for the mid-Holocene (MH)
(*MHcontrol–PIcontrol*) (a, b) and Last Interglacial (LIG) (*LIGcontrol–PIcontrol*) (c, d) conditions relative to the preindustrial (PI)
condition. The dotted area represents the region where 95% confidence interval was not achieved.**





For the MH case, the enhanced warming over high latitude regions compared with the PI conditions inferred from the paleoclimate proxy was also not simulated (Fig. 2a). Significant warming is observed (~2 K) only over the mid-latitude regions of the Atlantic Ocean. The warming over the Atlantic Ocean compared with the PI conditions is seen in every season (left panels in Fig. 3). Over the North American and Eurasian continents, the global annual mean surface air temperature decreased slightly (Fig. 2a). In these regions, the seasonal cycle of the surface air temperature strengthened compared with

the PI conditions (left panels in Fig. 3). In the JJA (June–Aug) and SON (Sep–Nov) seasons, the surface air temperature in some part of these regions increased by more than 1 K compared with the PI conditions, while it decreased by more than 1 K during the DJF (Dec–Feb) and MAM (Mar–May) seasons. As a result, the sea ice concentration during SON decreased compared with the PI conditions (Fig. 4). These changes are related to the change in insolation during the MH (Fig. 5a). Over the Arctic Ocean, the surface air temperature increases by ~1 K in SON (Fig. 5c), owing to the decreased sea ice

concentration (Fig. 4c).







**Figure 3: Surface air temperature anomaly relative to the preindustrial condition for MAM, JJA, SON, and DJF during the mid-Holocene (MH) (*MHcontrol–PIcontrol*) (left panels) and the Last Interglacial (LIG) (*LIGcontrol–PIcontrol*) (right panels). The dotted area represents the region where 95% confidence interval was not achieved.**





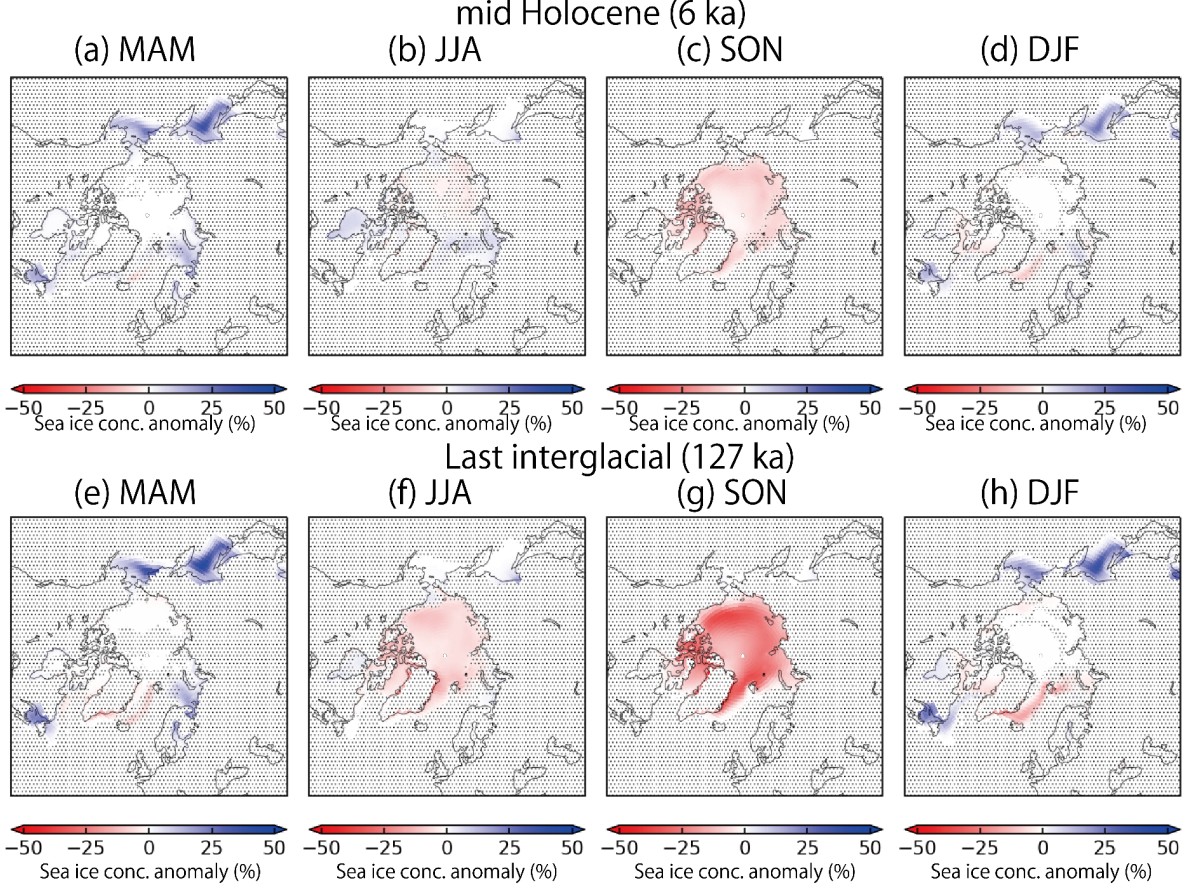

**Figure 4: Sea ice concentration anomaly around the Arctic Ocean relative to the preindustrial condition for MAM, JJA, SON, and DJF during (a–d) the mid-Holocene (MH) (*MHcontrol–PIcontrol*) and (e–h) the Last Interglacial (LIG) (*LIGcontrol–PIcontrol*). The dotted area represents the region where 95% confidence interval was not achieved.**






**Figure 5:** Top-of-atmosphere insolation anomaly and surface (2m) air temperature anomaly for the mid-Holocene (MH) condition (a and c, respectively) (*MHcontrol–PIcontrol*) and the Last Interglacial condition (b and d, respectively) (*LIGcontrol–PIcontrol*), relative to the preindustrial condition. The dotted areas in (c) and (d) represent the region where 95% confidence interval was not achieved.



For the LIG case, the overall patterns of the surface air temperature anomaly from the PI conditions were similar to the MH, but the signal tended to be stronger, which was similar to the previous study (Otto-Bliesner et al. 2021). Significant warming is observed over the mid-latitude regions of the Northern Hemisphere in the Atlantic Ocean and over the regions covering the Greenland Sea, Norwegian Sea, and Barents Sea (Fig. 2c). The warming is also observed in the central part of North America, the southern part of South America and Africa, the northern part of Australia, the western part of the Eurasian continent, and in part of the Southern Ocean. This significant warming in annual mean air temperature is primarily caused by the higher temperature in JJA and SON (right panels in Fig. 3), which is associated with the higher insolation anomaly (Fig. 5b). In SON, the surface air temperature over the Arctic Ocean increases by more than ~4 K compared with the PI conditions (Fig. 5d). As a result, the sea ice concentration during SON decreased over a wide area of the Arctic Ocean (Fig. 4g). Despite the warming in these regions, the global annual mean surface air temperature did not exceed the PI value.

The surface air temperature and sea ice distributions in the high latitude regions of the Southern Hemisphere during the MH and LIG are shown in Figs. 6 and 7, respectively. During the MH and LIG, the sea ice concentration anomaly in the Southern Ocean relative to the PI conditions exhibits a zonally asymmetric pattern in JJA and SON, associated with the surface air temperature anomaly (Figs. 6 and 7). The simulated pattern in JJA and SON was similar to the distribution anomaly owing to the direct influence of insolation simulated in the previous study (Pedersen et al. 2017). On the other hand, during the MH, the patterns of the sea ice concentration anomaly relative to PI in JJA differed from that in the previous study (Noda et al. 2017), which estimated that the sea ice concentration would increase in JJA in this region. In DJF and MAM, the sea ice concentration increased significantly in the case of the LIG. During the MH, the sea ice concentration showed a very small change in DJF, most of which was not significant (Fig. 7d). In MAM, the sea ice concentration increased significantly during the MH compared with the PI conditions (Fig. 7a).





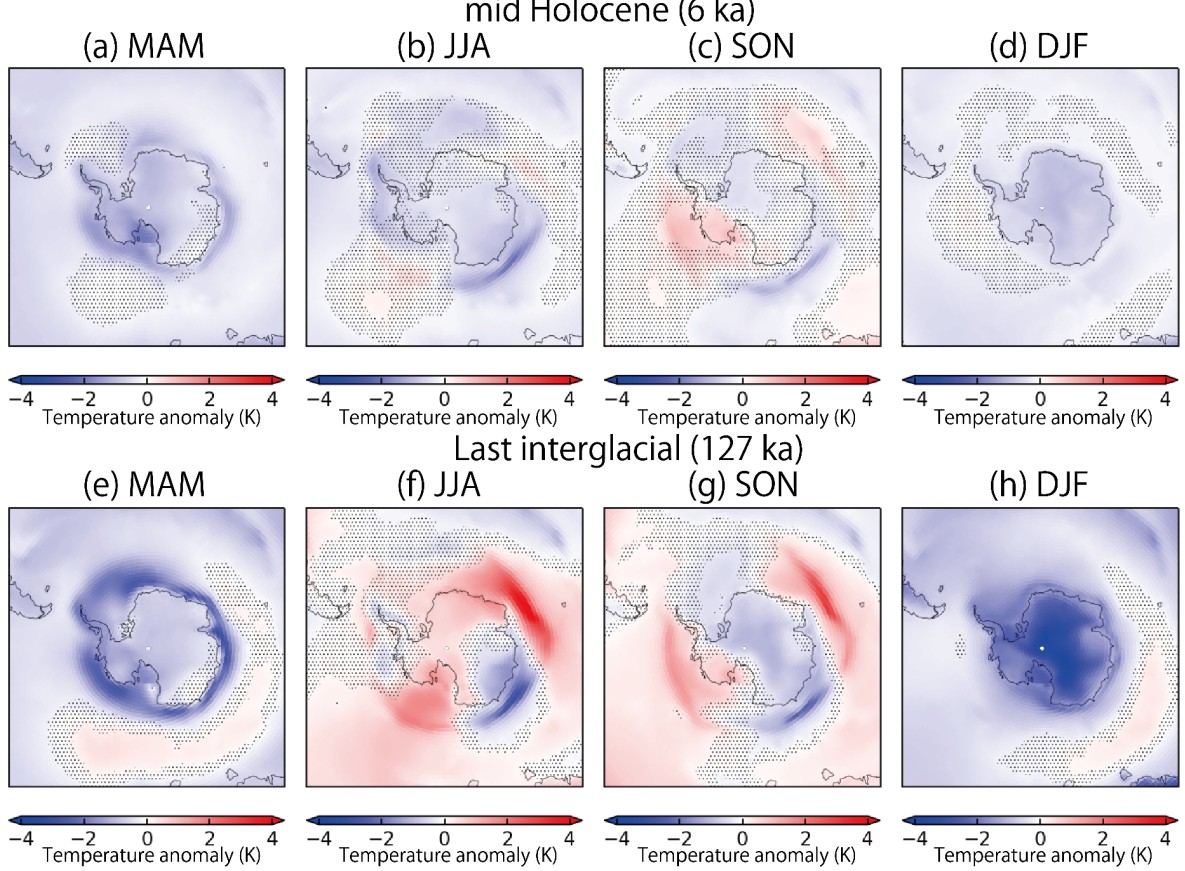

**Figure 6: Surface air temperature anomaly around the South Pole relative to the preindustrial condition for MAM, JJA, SON, and DJF during the mid-Holocene (MH) (*MHcontrol–PIcontrol*) (a–d) and the Last Interglacial (LIG) (*LIGcontrol–PIcontrol*) (e–h). The dotted area represents the region where 95% confidence interval was not achieved.**



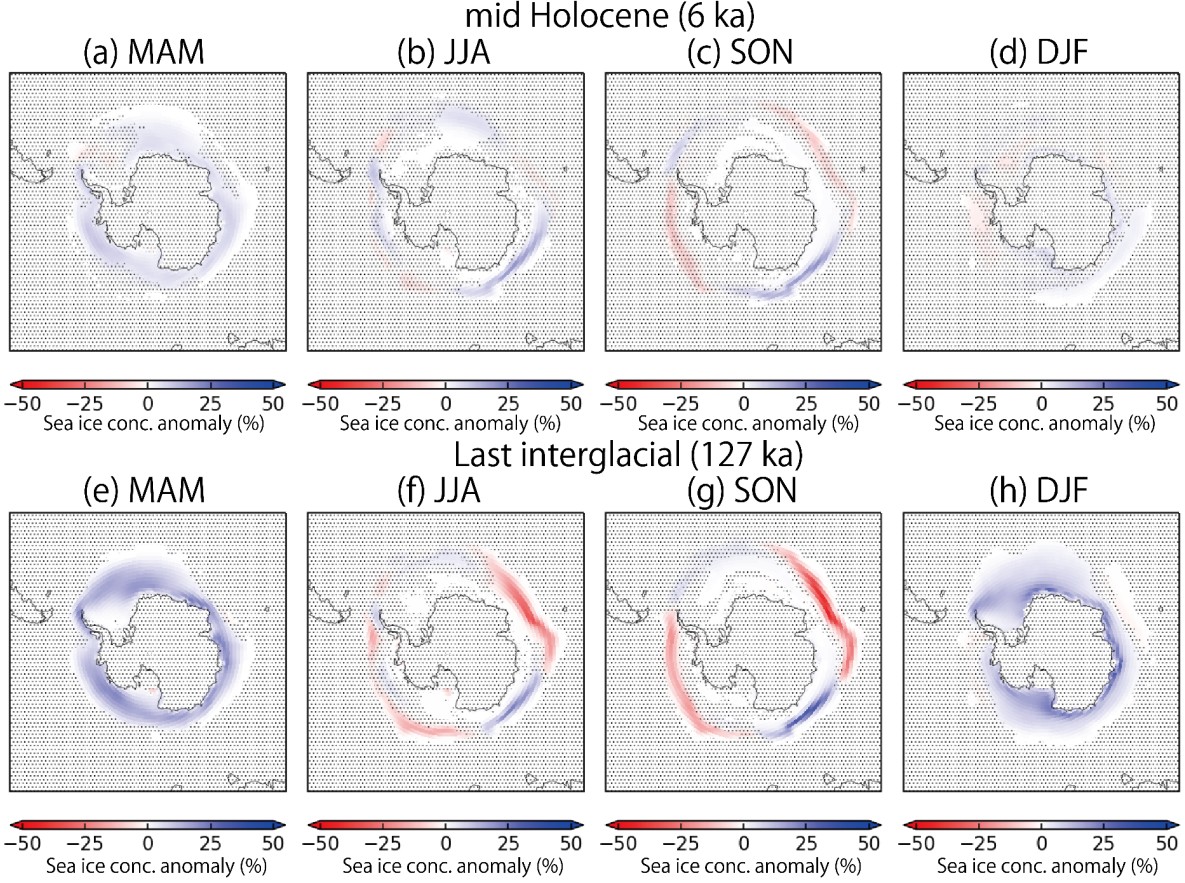

**Figure 7: Sea ice concentration anomaly around the South Pole relative to the preindustrial condition for MAM, JJA, SON, and DJF during the mid-Holocene (MH) (*MHcontrol–PIcontrol*) (a–d) and the Last Interglacial (LIG) (*LIGcontrol–PIcontrol*) (e–h). The dotted area represents the region where 95% confidence interval was not achieved.**



## 3.2 Response of the stratospheric ozone over Antarctica

The latitudinal and seasonal changes in the air temperature, and the ozone concentration in the upper stratosphere (3 hPa) are
shown in Fig. 8. In both the MH and LIG conditions, the changes in the temperature in the upper stratosphere correlate
positively with the changes in the top-of-atmosphere (TOA) insolation owing to the shortwave ozone heating (Figs. 5a, 5b,
8a, and 8b). The ozone concentration in the upper stratosphere correlates negatively to the insolation and temperature
anomalies. The decrease in the ozone concentration would mitigate the warming in the upper stratosphere, but this effect was
weaker than the effect of the changes in shortwave radiation during the MH and LIG. In the case of the MH conditions, the
ozone concentration anomaly turns negative during May and turns positive during January (Figs. 8c and 9c). This pattern is
broadly consistent with the results of the previous study using the former version of MRI-ESM under the MH conditions
(Noda et al. 2017). This would indicate a robust response of the ozone concentration to the different astronomical forcings
during the MH. For the case of the LIG conditions, the overall pattern of the seasonal response of the temperature and the
ozone concentration in the upper stratosphere was similar to that in the MH conditions, but the signal was stronger in the
LIG compared to that in the MH. The notable difference in the LIG is that the ozone concentration turns negative during
April and turns positive during October and November (Figs. 8d and 9d), following the insolation anomaly associated with
variations in the precession (Figs. 5b).





**Figure 8: Air temperature anomaly at a pressure level of 3 hPa (a–b), ozone concentration anomaly at a pressure level of 3 hPa (c–d), and eastward surface (10 m) wind speed anomaly (e–f) for the mid-Holocene (MH) condition (*MHcontrol–PIcontrol*) (a, c, e) and the Last Interglacial condition (*LIGcontrol–PIcontrol*) (b, d, f), relative to the preindustrial condition. The dotted area represents the region where 95% confidence interval was not achieved.**








**Figure 9: Vertical-seasonal plot of the air temperature anomaly (a, b) and ozone concentration (c, d) at high latitude regions of the**
**southern hemisphere (60–90°) and eastward wind speed anomaly at mid-latitude regions of the southern hemisphere (50–70°) (e, f)**
**relative to the preindustrial condition during the mid-Holocene (MH) (*MHcontrol–PIcontrol*) (a, c, e) and the Last Interglacial**
**(LIG) (*LIGcontrol–PIcontrol*) (b, d, f). The dotted area represents the region where 95% confidence interval was not achieved.**





The inverse correlation between the air temperature and ozone concentration is explained by the changes in the steady state of the chemical reactions in the Chapman cycle (Chapman 1930; Noda et al. 2017) because a higher temperature promotes the following ozone destruction reaction:

$$O_3 + O \rightarrow 2O_2. \tag{R1}$$

This would lead to the promotion of the other reactions in the Chapman cycle:

$$O_2 + hv \rightarrow 2O, \tag{R2}$$

$$O_2 + O + M \rightarrow O_3 + M, \tag{R3}$$

$$O_3 + hv \rightarrow O_2 + O. \tag{R4}$$

As a result, the ozone concentrations in the upper stratosphere around the southern polar region (90–60˚S) in the MH and LIG conditions are both higher than that in the PI conditions during the austral summer because the air temperature is lower than in the PI conditions owing to the weaker insolation (Fig. 9c and 9d). As a result, the ozone concentrations in the upper stratosphere are more than 0.1 ppm higher than the PI conditions during February and March for the MH conditions (Fig. 9c), which is slightly smaller than the peak value estimated in the previous study (Noda et al. 2017). In the LIG conditions, the increase in ozone concentration in the upper stratosphere starts during October and continues until April. This is because the insolation anomaly became negative during October, which resulted in the earlier onset of the positive anomaly of the ozone concentration compared with the case of the MH conditions (Fig. 5b). The positive anomaly of the ozone concentration in the upper stratosphere during the LIG is much stronger than in the MH case, exceeding ~0.3 ppm during January (Fig. 9d).

The positive ozone anomaly in the upper stratosphere during the austral summer is transported into the lower stratosphere during the austral autumn and winter where the lifetime of ozone is relatively long owing to the polar night (Noda et al. 2017). As a result, the positive ozone anomaly persists in the lower stratosphere (~100 hPa) until March of the following year for the MH case (Fig. 9c). For the LIG case, the positive anomaly was stronger than in the MH, and it persists in the lower stratosphere throughout the year (Fig. 9d). In this case, the positive anomaly was over 0.05 ppm throughout the year. This may have contributed to warming the lower stratosphere, especially from February to June, as seen when the contribution to air temperature from ozone changes is shown (Fig. 10b).





**Figure 10: Vertical-seasonal plot of the ozone-induced air temperature anomaly (a, b) and ozone-induced eastward wind speed anomaly at mid-latitude regions of the southern hemisphere (50–70˚) (c, d) relative to the preindustrial condition during the mid-Holocene (MH) (*MHcontrol–PIcontrol–MHnoChem+PInoChem*) (a, c) and the Last Interglacial (LIG) (*LIGcontrol–PIcontrol–LIGnoChem+PInoChem*) (b, d). The dotted area represents the region where 95% confidence interval was not achieved.**



During the austral winter to spring, the ozone concentration anomaly in the upper stratosphere decreases owing to
the high air temperature (Fig. 9a–9d). The negative anomaly drops down to ~0.2 and ~0.3 ppm for the case of the MH and
LIG, respectively. Concurrently, the strength of the southern westerly jet increases dramatically during the austral winter and
spring (Fig. 9e and 9f). The wind speed increases to more than 8 m s$^{-1}$ during both the MH and LIG. This intensification of
the southern westerly jet works opposite to the effect from changes in the ozone distribution inferred from the previous study
(Noda et al. 2017). The westerly jet intensity is related to the meridional air temperature gradient in the stratosphere at the
mid-latitude regions, which is attributed to the insolation anomaly. Notably, the intensification of the westerly jet is
accelerated by the change in the atmospheric ozone in both the MH and LIG (Fig. 10c and 10d), even though the ozone-
related positive air temperature anomaly indicates a smaller meridional temperature gradient during the austral winter and
spring (Fig. 10a and 10b).

The positive wind anomaly in the stratosphere during the austral winter propagates into the troposphere in the
austral spring (Figs. 8e, 8f, 9e, and 9f). The seasonal changes in the near-surface wind speed (10 m) anomaly are shown in
Fig. 11. During SON, the near-surface wind speed increases over a wide area of the ~40–60˚S region in both the MH and
LIG (Fig. 11b, 11c, 11f, and 11g). During DJF and MAM, the near-surface wind speed decreased compared with the PI
conditions over a wide area of the ~40–60˚S region in the MH and LIG (Fig. 11a, 11b, 11e, and 11f). The changes in the
near-surface wind speed are considered to have affected the climate via changing the surface Ekman transport (Noda et al.
2017), which may further modify the sea ice distribution and surface air temperature. However, the response of the surface
air temperature was rather asymmetric zonally in JJA and SON during the LIG (Fig. 6f and 6g). In addition, the ozone-
related change in the surface air temperature summarised in Fig. 12 exhibits the asymmetric effect in JJA and SON,
especially in the LIG, while regions with significant signals are limited (Fig. 12f and 12g). These results indicate that the
change in the atmospheric ozone affects the surface air temperature only regionally, and the ozone-induced near-surface
wind and sea ice distribution changes inferred in Noda et al. (2017) do not operate in our results. As a result, the zonally
averaged surface air temperature anomaly shown in Fig. 13 indicates that the impact of the atmospheric ozone changes in the
southern zonally averaged climate was small in both the MH and LIG. This is contradictory to the results shown by Noda et
al. (2017), which suggests a warming in the Southern Hemisphere in the MH. Rather, our results show a cooling of the
zonally averaged temperature at the high latitude regions of the Northern Hemisphere. This cooling owing to atmospheric
ozone can be seen in any season, and is specifically strong during SON and DJF in the Barents Sea (Fig. 14e–14h), possibly
reflecting a mechanism associated with sea ice dynamics. In summary, these results indicate that the changes in the
stratospheric ozone may affect the surface climate at least regionally around the poles.



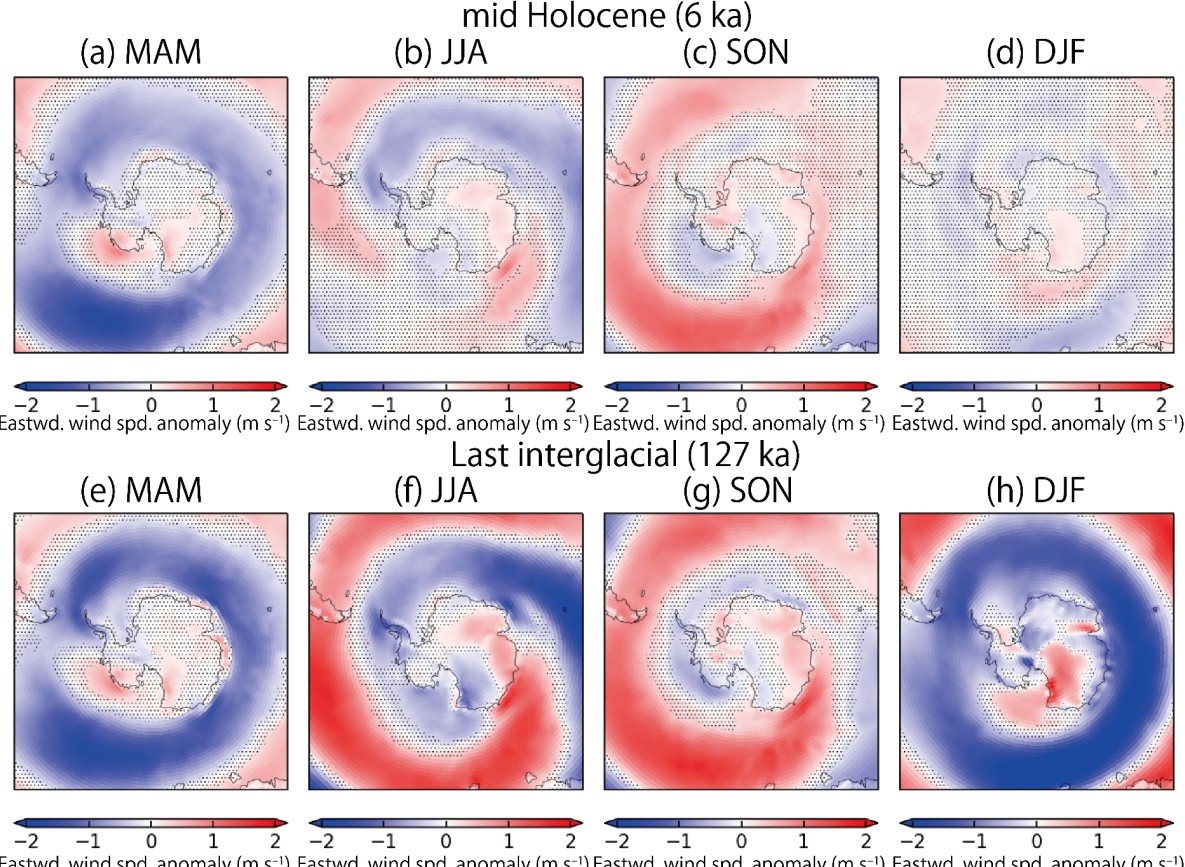

Figure 11: **Eastward near-surface (10 m) wind anomaly around the South Pole relative to the preindustrial condition for MAM, JJA, SON, and DJF during the mid-Holocene (MH)** (*MHcontrol–PIcontrol*) **(a–d) and the Last Interglacial (LIG)** (*LIGcontrol–PIcontrol*) **(e–h). The dotted area represents the region where 95% confidence interval was not achieved.**



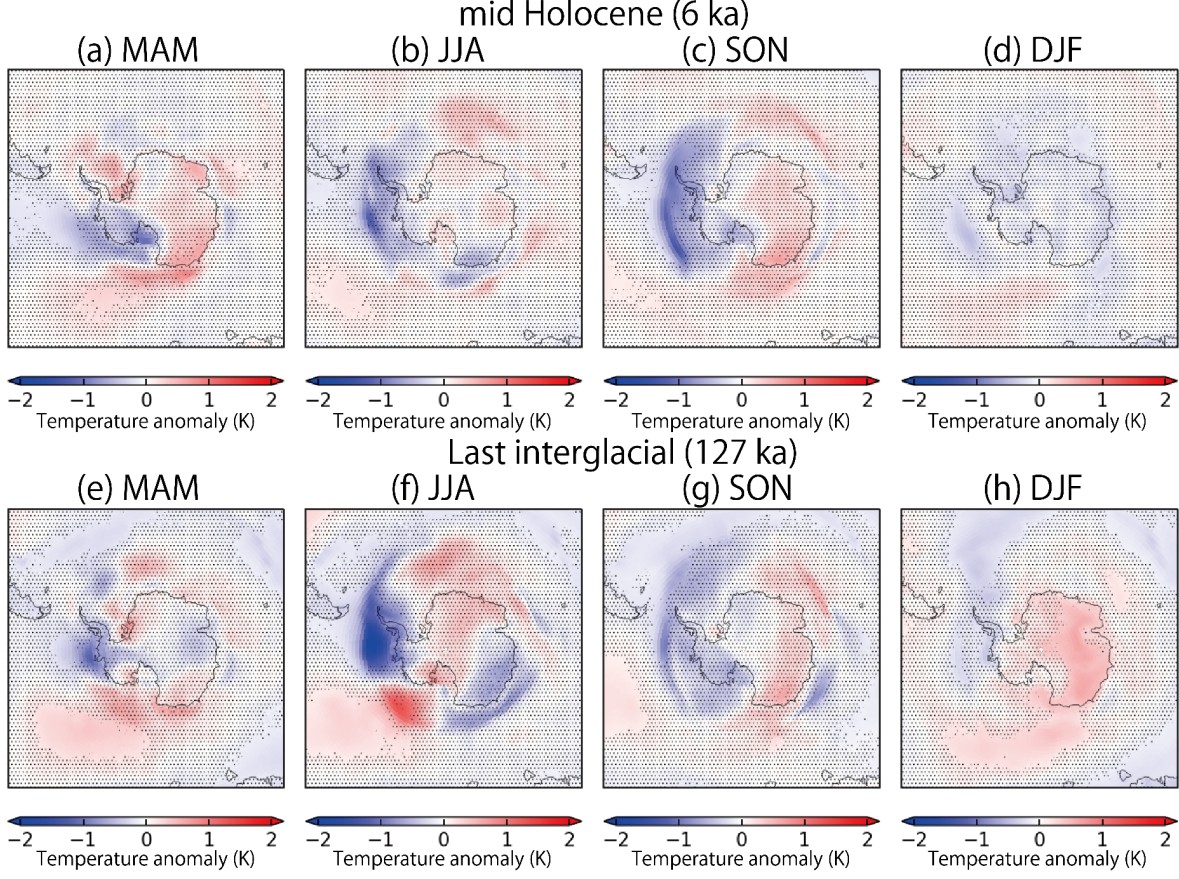

**Figure 12: Ozone-induced surface air temperature anomaly around the South Pole relative to the preindustrial condition for MAM, JJA, SON, and DFJ during the mid-Holocene (MH) (*MHcontrol–PIcontrol–MHnoChem+PInoChem*) (a–d) and the Last Interglacial (LIG) (*LIGcontrol–PIcontrol–LIGnoChem+PInoChem*) (e–h). The dotted area represents the region where 95% confidence interval was not achieved.**





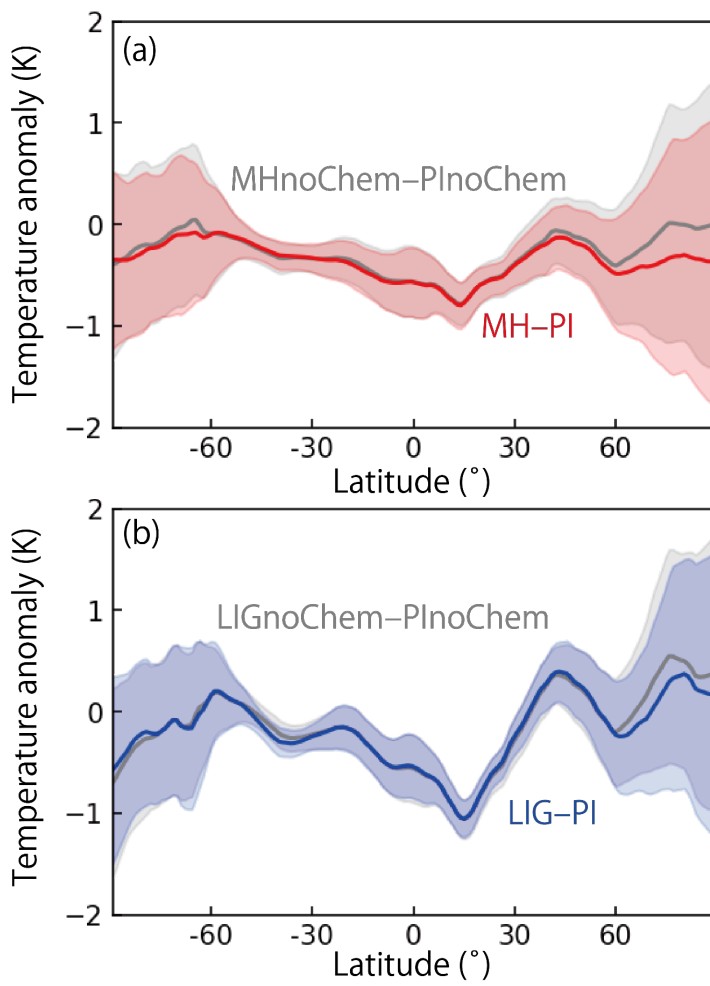


**Figure 13: Zonally averaged annual mean temperature anomaly for (a) mid-Holocene (MH) and (b) Last Interglacial (LIG) conditions compared with the preindustrial (PI) condition. The red and blue lines represent the zonally mean surface air temperature anomaly for the MH and LIG experiments, respectively (*MHcontrol–PIcontrol* and *LIGcontrol–PIcontrol*, respectively). The grey lines represent the zonally mean surface air temperature anomaly for the MH and LIG experiments estimated using an atmospheric ozone distribution in PI, respectively (*MHnoChem–PInoChem* and *LIGnoChem–PInoChem*, respectively). The hatched regions represent the intervals of one sigma.**




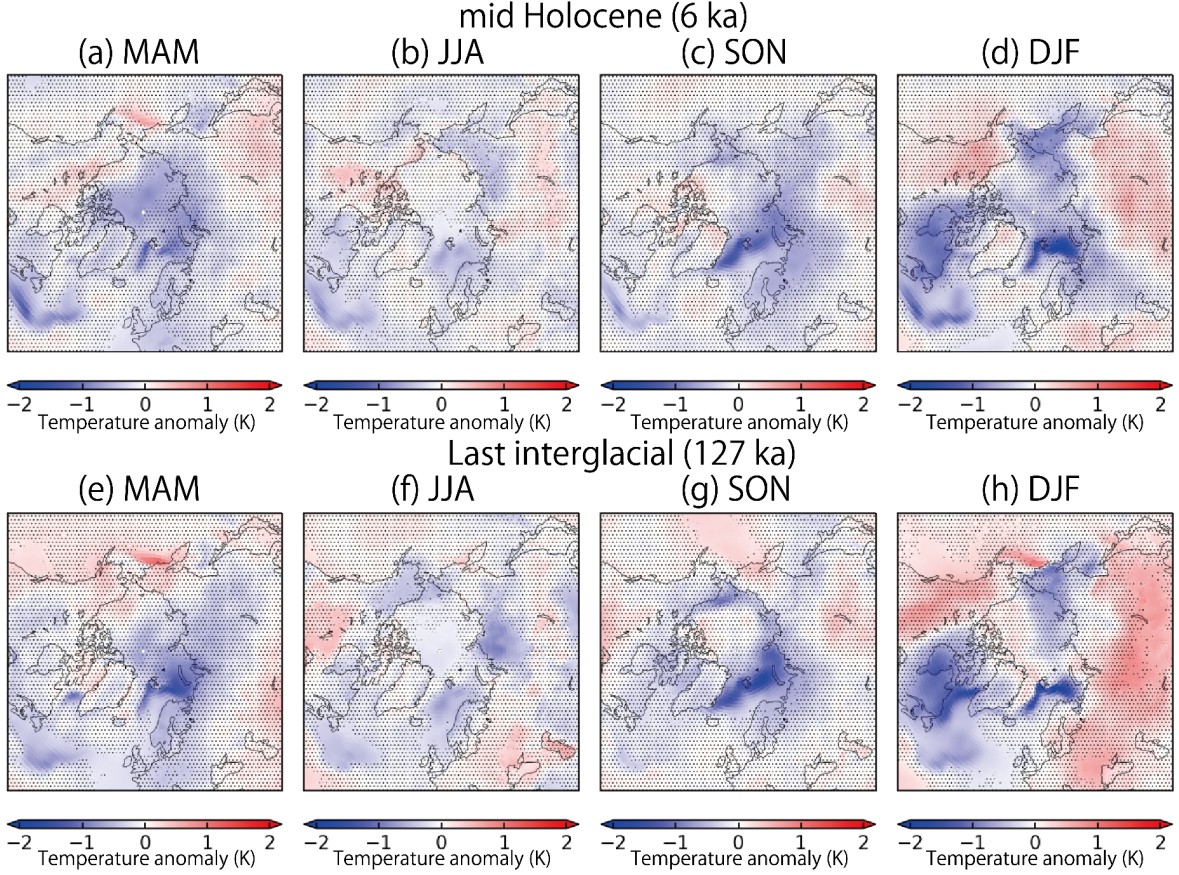

**Figure 14: Figure 14. Ozone-induced surface air temperature anomaly around the North Pole relative to the preindustrial condition for MAM, JJA, SON, and DJF during the mid-Holocene (MH) (*MHcontrol–PIcontrol–MHnoChem+PInoChem*) (a–d) and the Last Interglacial (LIG) (*LIGcontrol–PIcontrol–LIGnoChem+PInoChem*) (e–h). The dotted area represents the region where 95% confidence interval was not achieved.**

## 4 Discussion

In this study, we investigated the climate and ozone distribution during the MH and LIG using MRI-ESM2.0. For both the MH and LIG, our model did not simulate warmer climates both during than that in the PI period, which is similar to the situations of many other climate models simulating the MH (Liu et al. 2014; Brierley et al. 2020) and the LIG (Lunt et al. 2013; Masson-Delmotte et al. 2013; Otto-Bliesner et al. 2013; Otto-Bliesner et al. 2020; Williams et al. 2020; Otto-Bliesner et al. 2021; Zhang et al. 2021). The colder temperature is attributed primarily to the cold temperature in the Sahara and Sahel regions in North Africa as observed in many other models (Lunt et al. 2013; Brierley et al. 2020; Otto-Bliesner et al. 2021), which would be associated with the formation of clouds in these regions. It has been discussed from the paleoclimate records





that much of the present Sahara Desert was covered by grassland during the MH and LIG (Hoelzmann et al. 1998; Bartlein et al. 2011; Tarasov et al. 2013; Hély et al. 2014; Hoogakker et al. 2016; Harrison 2017; Phelps et al. 2020). This would be associated with the bistable characteristics of the vegetation cover in this region and an abrupt transition to a vegetated state

(Brovkin et al. 1998; Hopcroft and Valdes 2021, 2022). A study using a climate model inferred that this may contribute to a warmer global annual mean surface temperature during the MH (Thompson et al. 2022). In addition, a simulation with decreased dust emission from the Sahara Desert assuming the greening of Sahara indicates that the reduced dust emission may help warm the surface (Liu et al. 2018). These results indicate that a reproduction of the greening of the Sahara Desert would be key for achieving a warm climate during the MH and LIG in climate models. Recently, some studies have

reproduced the greening of the Sahara Desert using climate models coupled with a dynamic vegetation model (Lu et al. 2018; Hopcroft and Valdes 2021; Hopcroft et al. 2021; Duque-Villegas et al. 2022; Specht et al. 2024), while many other climate models failed to reconstruct the desert (Joussaume et al. 1999; O'ishi and Abe-Ouchi 2011; Harrison et al. 2015). To sustain the greening of the Sahara Desert, feedback that helps sustain the precipitation over this region may be necessary, including the formation of large lakes (Contoux et al. 2013; Li et al. 2023). Indeed, a recent model that dynamically

simulates the vegetation and lake distributions indicates the greening of the Sahara (Specht et al. 2024). Currently, the land surface type in the atmospheric model of MRI-ESM2.0 has not changed dynamically in accordance with the climate changes. Future studies considering these processes in Earth system models would be highly desirable for understanding the dynamics of the greening of the Sahara Desert and the warm climate during the MH and LIG.

    In this study, we showed that the southern westerly jet would strengthen during the austral winter and autumn

during the MH and LIG owing to the different astronomical forcing. In the previous study, it was pointed out that the changes in the stratospheric ozone distribution would weaken the westerly jet during the austral winter (Noda et al. 2017). This was considered to be associated with the positive ozone anomaly in the lower stratosphere using the previous version of MRI-ESM (MRI-ESM1) (Noda et al. 2017). However, our results show the strengthening westerly jet, an opposite response to that reported by Noda et al. (2017), even though the response of the ozone was similar to that by Noda et al. (2017). This

is because the change in the astronomical forcing increases the temperature in low latitude regions more than in the high latitude regions of the Southern Hemisphere, increasing the meridional temperature gradient in the stratosphere. This means that the different astronomical forcing is an important driver of the intensification of the southern westerly jet, while the variations in the ozone layer during the MH and LIG also amplify this effect. As a result, the responses of the sea ice distribution and the surface air temperature differed from that reported by Noda et al. (2017), exhibiting zonally asymmetric

characteristics. Our results exhibit a minor impact of the atmospheric ozone on global mean climate states, although it may impact the local surface air temperature seasonally around Antarctica. For further quantitative constraints on the role of the stratospheric ozone in paleoclimates, it should be investigated using multiple Earth system models with interactive ozone.

    While the reconstruction of stratospheric ozone in the past is not available, the tropospheric ozone burden during the LIG was estimated based on a record of the clumped isotope composition of $O_2$ in the East Antarctic ice core (Yan et al.

2022), indicating a reduction in the tropospheric ozone burden by nearly 9 % compared with the PI conditions. It has been

inferred that the dispersal of modern humans had not yet occurred during the LIG (Liu et al. 2015; Malaspinas et al. 2016; Groucutt et al. 2018), which makes the LIG an ideal period for reconstructing the tropospheric ozone under a smaller impact from the influence of humans (Yan et al. 2022). The important factor affecting the amount of tropospheric ozone is the concentration of methane. Ice core records indicate that the atmospheric methane level has increased since the MH compared

with the LIG (Spahni et al. 2005; Singarayer et al. 2011), which would be associated with rice cultivation (Ruddiman 2003; Ruddiman et al. 2008) and/or natural wetland emissions (Schmidt and Shindell 2004; Sowers 2010; Singarayer et al. 2011). The tropospheric ozone is also affected by biomass burning associated with wildfire events, which produces methane and $NO_x$ and contributes to consuming the atmospheric ozone (Ward et al. 2012). The magnitude of wildfire events is determined by the complex interrelationship between climate, vegetation activities, and early human activities. However, a dynamic

methane cycle and wildfire activity have not been considered in MRI-ESM2.0. Simulations of the last glacial cycle using a fully coupled Earth system model including the atmosphere, ocean, aerosol, ozone chemistry, and vegetation cycle would be ideal for understanding the variations in the tropospheric ozone distribution during this period.

## 5 Conclusions

This study investigated the climate and atmospheric ozone simulated by MRI-ESM2.0. The response of the atmospheric

ozone during the MH was similar to the results of the previous study. The pattern of the atmospheric ozone response during the LIG was similar to that during the MH, but the intensity of the variation was stronger than in the MH. The southern westerly jet intensified compared with the PI conditions during the austral winter and spring owing to the effect from the astronomical forcing, which was further amplified by the variations in the atmospheric ozone. The change in the atmospheric ozone affects the surface air temperature only regionally around the poles and its impact on the zonal mean surface air

temperature is small.

### Code and data availability

The source code of MRI-ESM2.0 is the property of the Meteorological Research Institute, Japan Meteorological Agency, and not available to the general public. It will be available upon reasonable request under a collaborative framework with Meteorological Research Institute. Data are available from the corresponding author upon request.

### Author Contributions

Conceptualization: YW; Data Curation: YW; Methodology: YW, KY, MD; Supervision: MD; Visualization: YW; Writing – original draft: YW; Writing – review & editing: YW, MD, KY



## Competing Interests

The authors declare that they have no conflict of interest.

**Acknowledgements**

We appreciate Y. Yokoyama, A. Abe-Ouchi, and R. O'ishi for fruitful discussion.

**Financial support**

This work was supported by JSPS KAKENHI grant numbers 24H02193 and 20K04070.

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

E3