# Peer review of "Climate and stratospheric ozone during the mid-Holocene and Last Interglacial simulated by MRI-ESM2.0"

_EGUsphere, 2024_

## Referee Comment (RC1)

**Climate and stratospheric ozone during the mid-Holocene and Last Interglacial simulated by MRI-ESM2.0**

By Yasuto Watanabe, Makoto Deushi, and Kohei Yoshida

**Summary:** Watanabe et al. utilize MRI-ESM2.0, an Earth system model with an ozone module, to simulate climate and atmospheric ozone changes for the preindustrial (PI), mid-Holocene (MH), and Last Interglacial (LIG). Their study explores ozone-climate feedbacks by selectively enabling and disabling the ozone-chemistry module, with a particular focus on high-latitude regions. While their results indicate that stratospheric ozone changes can influence polar surface air temperatures, they suggest a limited impact on global mean temperature. The study extends previous work by examining both MH and LIG. It raises questions about model dependency and the need for further multi-model comparisons.

The manuscript requires improvement in several areas, including the need for clearer differentiation between the sizes and sources of human-induced and natural ozone changes (i.e. quantification of ozone changes in ppm), and much more careful and robust discussion of sea ice state errors and their implications. Overall the study requires that it includes findings from previous research on MH/LIG polar climate changes, particularly regarding Arctic sea ice extent and its role in driving feedbacks. Further smaller clarifications are also required on spin-up model-specific biases.

**Line-by-line comments:**

L9: Remove: "However, understanding the role of changes in stratospheric ozone during past warm interglacial periods is limited to MH conditions," since work suggests that previous understanding was incorrect.

L13: Reverse clauses for better sentence construction: "We show that while ozone feedbacks may affect surface air temperature regionally, impacts on the zonal mean surface air temperature are small."

L14: Add a sentence explaining that these results represent an update on previous findings or that further work using more models is needed to determine whether this indicates model dependency.

L14-15: Remove or rewrite the last sentence to reflect the previous comment.

L16: Change "is expected" to "can."

L24-25: Split land-ocean versus sea ice-ocean feedbacks (Arctic versus other land) and rewrite the sentence clearly.

L27: Remove the sentence on ice volume/GMSL changes. The authors do not address this here, and it adds nothing.

L29-L34: These lines confuse the MH and LIG and add nothing. They can be removed.

L29-L34: Instead, provide a clear description of what is known about MH and LIG polar (sea ice and polar ocean) changes, particularly sea ice. Refer to:

- Gao, Qinggang, et al. (2025) *Assessment of the southern polar and subpolar warming in the PMIP4 Last Interglacial simulations using paleoclimate data syntheses*. Climate of the Past, 21. 10.5194/cp-21-419-2025

- Sime, Louise C., et al. (2025) *More modest peak temperatures during the Last Interglacial for both Greenland and Antarctica suggested by multi-model isotope simulations.* Climate of the Past [in review]. 10.5194/egusphere-2025-288

- Chadwick, Matthew, et al. (2023) *Model-data comparison of Antarctic winter sea-ice extent and Southern Ocean sea-surface temperatures during Marine Isotope Stage 5e.* Paleoceanography and Paleoclimatology, 38(11). 10.1029/2022PA004600

- Sime, Louise C., et al. (2023) *Summer surface air temperature proxies point to near-sea-ice-free conditions in the Arctic at 127 ka.* Climate of the Past, 19. 10.5194/cp-19-883-2023

- Diamond, Rachel, et al. (2021) *The contribution of melt ponds to enhanced Arctic sea-ice melt during the Last Interglacial.* The Cryosphere, 15(16). 10.5194/tc-15-5099-2021

- Kageyama, Masa, et al. (2021) *A multi-model CMIP6-PMIP4 study of Arctic sea ice at 127 ka: Sea ice data compilation and model differences.* Climate of the Past, 17(26). 10.5194/cp-17-37-2021

- Guarino, Maria Vittoria, et al. (2020) *Sea-ice-free Arctic during the Last Interglacial supports fast future loss.* Nature Climate Change, 10. 10.1038/s41558-020-0865-2

- Williams, Charles J.R., et al. (2020) *CMIP6/PMIP4 simulations of the mid-Holocene and Last Interglacial using HadGEM3: comparison to the pre-industrial era, previous model versions, and proxy data.* Climate of the Past, 16(22). 10.5194/cp-16-1429-2020

When constructing a paragraph about polar changes for the LIG, the *Sime et al. (2025)* reference summarizes much of what is required.

L40: Change sentences to clarify that the focus is on ozone-climate feedbacks. Change to: "One possible factor that can affect the high latitudes is stratospheric ozone-climate feedbacks (Thompson and Wallace, 2000; Noda et al., 2017)."

L41: Rewrite to clarify the difference between human-generated ozone changes and the ozone-climate feedbacks being investigated.

L47: Add numbers to show the size of the effects: ozone changes in ppm for present-day (human-induced) versus possible ppm changes for the MH or other past climates due to ozone-climate feedbacks.

L49: Again, ensure the previous estimated response size in ppm from Noda et al. is explicitly stated.

L53-66: Clearly spell out that the two objectives of this study are:

1. Testing whether a newer model yields the same results as the previous MH study.

2. Extending the work on interglacials from the MH to both the MH and LIG.

Table 1 / Methods Section: The spin-up process is unclear. Is everything initiated from the same well-spun-up PI? Add comments on the usual spin-up duration (>250 years) and its importance for polar regions. Refer to Kageyama et al. (2021) for comments on this.

L110-L115: Check whether this result is dependent on calendar adjustments and comment or adjust accordingly.

Figures 4, 7, and SIC-related figures: These figures should also show the actual PI and Interglacial SIE or SIC (add a 15% SIC line for each climate to each figure), not just anomalies. See *Kageyama and Sime* papers for why sea ice states/errors (in the PI and MH/LIG) are critical for determining SIC-climate changes (not just anomalies). Discuss any PI or MH/LIG sea ice state errors and their likely impacts.

L133: Add more appropriate references and comments based on the MH/LIG sea ice and polar change papers listed above.

L133-151: This section is structured backward. It is primarily direct insolation impacts on Arctic sea ice that reduce the SIE (SIC), leading to warming and subsequent climate changes. See *Diamond et al., Kageyama et al.,* and *Sime et al.* for clarification. Rewrite these paragraphs accordingly.

L168-171: Sentence is unclear—rewrite for clarity.

L171-175: Similar to the previous comment. The mixed tenses (previous interglacial times vs. previous Noda et al. results) make these lines difficult to parse. Separate:

- Climate-to-ozone feedback processes.

- Ozone-to-climate processes.

- MH/LIG simulation changes.

- Changes in the representation of climate-ozone-climate feedbacks.

- Differences between the Noda et al. results (previous model) and the new findings.

L240: Change "operate" to "occur."

L243: "This contradicts the results shown by Noda et al. (2017), which suggest a warming in the Southern Hemisphere during the MH." This difference should be clearly stated in the abstract.

L245: Change "any season" to "all seasons" and "specifically" to "particularly."

L247: Contextualize the size of the changes relative to previously identified LIG and MH sea ice changes (Guarino, Sime, Chadwick, Gao, and Kageyama et al.).

Figures 10 and 13: Show ozone-dependent impacts in K for pressures and latitudes (e.g. ~0.25K in the high Arctic). Explicitly state the magnitude of these numbers in the abstract and conclusions.

L276: Spell out "mean annual" and "globally"—for example, large seasonal Arctic changes exist.

L281-L298: Since the study focuses on ozone impacts in polar/high-latitude regions, remove the discussion on global mean temperature and non-polar changes. Instead, discuss whether either model version accurately captures known MH/LIG sea ice and surface polar ocean changes and how that affects the climate-ozone-climate feedbacks.

L299-L312: Clarify which aspects are model-specific (e.g., climate biases affecting interpretation) and state the headline results.

L313-L327: This paragraph is difficult to parse. If the argument is that further chemistry should be included in the model, first state which chemistry is currently missing, then explain why this could be important for MH, LIG, or another past climate interval.

---

## Referee Comment (RC2)

A review of

**Climate and stratospheric ozone during the mid-Holocene
and Last Interglacial simulated by MRI-ESM2.0**

(by Watanabe et al.)

**Review.**    This paper explores the effect of stratospheric ozone changes – if any – on the climate of the mid-Holocene (MH) and the Last Interglacial (LIG). The question is well posed, and most interesting. To answer it the authors have performed an excellent set of model runs: 3 epochs (1850 PI control, MH and LIG) and, for each epoch, 2 runs (with PI ozone and with interactive ozone). So, they should be able to answer the question clearly.

Unfortunately the manuscript, in its current form, is really a mess. The authors waste 7 figures (each with many panels) discussing all manner of secondary considerations, and only show an ozone field for the first time in Figure 8. So, the narrative is completely backwards. If one is trying to tell the impact of ozone changes, one should start by showing the ozone changes. But this need to be done properly. Why are we shown ozone at the 3 hPa level in Figure 8? Is that were the ozone layer is? I would imagine the readers want to see ozone at 50 or 70 hPa. What about a latitude/height map of ozone changes in the ML and LIG? Or again, how about showing a lat/lon map of total column ozone (TCO)? Is the ozone layer thicker or thinner than under PI forcings? By how many Dobson units? I have no idea what the answer is (as I have not run the models), but none of this is shown in the paper. Again: the paper needs to start with 2 or 3 well chosen figures telling us what ozone looks like in the MH and LIG, and how it differs from the PI control.

Next, the key results are at the very end of the paper, in Figures 12, 13 and 14: these show that ozone changes in the ML and LIG have *basically no statistically significant impact* on surface climate. So, why are the most important figures left at the end of the paper? And why are the authors not stating clearly that the effect of interactive on surface temperature are minuscule? And what about precipitation (which is not show)? I suspect ozone is also irrelevant for the that. In my mind that should be the key point of the paper: ozone changes in the ML and LIG don't matter. It is a null result, but null results are *very much* worth publishing. In all honesty, I am not surprised that ozone changes make no difference: this is because I suspect these changes are small. It takes something like an ozone hole over the South Pole (as we have seen in the late 20th century) to make a substantial climate impact. Hence the key figure the readers need to see: *how big* are the ozone changes in the ML and LIG compared to those caused by CFCs?

**Recommendation.**    The paper – in its present form – should be rejected. However, the authors should be strongly encouraged to resubmit. They have a nice set of runs, and a very clean story to tell: the ozone changes in the ML and LIG are small, and therefore they make little difference for the surface climate. Such a paper is easy to write, as the key points can be made with a few simple figures, and no complicated mechanisms needed to be invoked. It will be a nice contribution to the literature. I look forward to it.

---

## Author Comment (AC1)

Summary: Watanabe et al. utilize MRI-ESM2.0, an Earth system model with an ozone module, to simulate climate and atmospheric ozone changes for the preindustrial (PI), mid-Holocene (MH), and Last Interglacial (LIG). Their study explores ozone-climate feedbacks by selectively enabling and disabling the ozone-chemistry module, with a particular focus on high-latitude regions. While their results indicate that stratospheric ozone changes can influence polar surface air temperatures, they suggest a limited impact on global mean temperature. The study extends previous work by examining both MH and LIG. It raises questions about model dependency and the need for further multi-model comparisons. The manuscript requires improvement in several areas, including the need for clearer differentiation between the sizes and sources of human-induced and natural ozone changes (i.e. quantification of ozone changes in ppm), and much more careful and robust discussion of sea ice state errors and their implications. Overall the study requires that it includes findings from previous research on MH/LIG polar climate changes, particularly regarding Arctic sea ice extent and its role in driving feedbacks. Further smaller clarifications are also required on spinup model-specific biases.

We appreciate the reviewer for the thoughtful review and providing many valuable comments. In the course of revision, we will clarify that our model does not consider the human-induced effect on tropospheric ozone in numerical experiments conducted in this study. We will further clarify the distribution of the sea ice during the PI, MH, and LIG and also compare our sea-ice distribution to the ones estimated by other climate models. We will also enrich our mention of the abundance and distribution of atmospheric ozone during the MH and LIG. We will take all these comments and suggestions into account in the process of revision.

Line-by-line comments:

L9: Remove: "However, understanding the role of changes in stratospheric ozone during past warm interglacial periods is limited to MH conditions," since work suggests that previous understanding was incorrect.

We agree with the reviewer. Instead of removing the sentence, we will revise the sentence as follows:

*"However, little is known about the role of changes in the stratospheric ozone during past warm interglacial periods."*

L13: Reverse clauses for better sentence construction: "We show that while ozone feedbacks may affect surface air temperature regionally, impacts on the zonal mean surface air temperature are small."

We will rewrite the sentence as suggested.

L14: Add a sentence explaining that these results represent an update on previous findings or that further work using more models is needed to determine whether this indicates model dependency.

Following the suggestion, we will add sentence as follows:

*"These results are the opposite of the previous finding that implies the importance of ozone in southern hemisphere climates, indicating the need to determine whether this indicates model dependency."*

L14-15: Remove or rewrite the last sentence to reflect the previous comment.

Following the suggestion, we will remove the last sentence.

L16: Change "is expected" to "can."

We will change the expression as suggested.

L24-25: Split land-ocean versus sea ice-ocean feedbacks (Arctic versus other land) and rewrite the sentence clearly.
We will rewrote the sentence as follows:

> *"The warming is especially enhanced over high-latitude regions, possibly reflecting the amplification by feedback mechanisms such as sea-ice changes over the Arctic Ocean and vegetation feedback over land areas of the Northern Hemisphere…"*

L27: Remove the sentence on ice volume/GMSL changes. The authors do not address this here, and it adds nothing.
We agree with the reviewer. We will remove the sentence in the revised manuscript.

L29-L34: These lines confuse the MH and LIG and add nothing. They can be removed.
We will rewrite the sentence so that the discrepancy between temperature reconstruction and climate model exists in both MH and LIG, as follows:

> *"However, many climate models do not simulate the higher global annual mean surface air temperature inferred from the paleoclimate proxy during the MH and LIG (Masson-Delmotte et al. 2013; Otto-Bliesner et al., 2013; Liu et al. 2014; Brierley et al. 2020; Kaufman and Broadman 2023). The cause of this discrepancy has been vigorously debated (Liu et al. 2014, 2018; Hopcroft and Valdes 2019; Park et al. 2019; Bova et al. 2021; Zhang and Chen 2021; Thompson et al. 2022; Laepple et al. 2022; Kaufman and Broadman 2023), indicating that high-latitude processes over land and ocean are critical for further understanding the climate warming during the past interglacials."*

L29-L34: Instead, provide a clear description of what is known about MH and LIG polar (sea ice and polar ocean) changes, particularly sea ice. Refer to:
- Gao, Qinggang, et al. (2025) Assessment of the southern polar and subpolar warming in the PMIP4 Last Interglacial simulations using paleoclimate data syntheses. Climate of the Past, 21. 10.5194/cp-21-419-2025
- Sime, Louise C., et al. (2025) More modest peak temperatures during the Last Interglacial for both Greenland and Antarctica suggested by multi-model isotope simulations. Climate of the Past [in review]. 10.5194/egusphere-2025-288
- Chadwick, Matthew, et al. (2023) Model-data comparison of Antarctic winter sea-ice extent and Southern Ocean sea-surface temperatures during Marine Isotope Stage 5e. Paleoceanography and Paleoclimatology, 38(11). 10.1029/2022PA004600
- Sime, Louise C., et al. (2023) Summer surface air temperature proxies point to near-seaice-free conditions in the Arctic at 127 ka. Climate of the Past, 19. 10.5194/cp-19-883-2023
- Diamond, Rachel, et al. (2021) The contribution of melt ponds to enhanced Arctic seaice melt during the Last Interglacial. The Cryosphere, 15(16). 10.5194/tc-15-5099-2021
- Kageyama, Masa, et al. (2021) A multi-model CMIP6-PMIP4 study of Arctic sea ice at 127 ka: Sea ice data compilation and model differences. Climate of the Past, 17(26). 10.5194/cp-17-37-2021
- Guarino, Maria Vittoria, et al. (2020) Sea-ice-free Arctic during the Last Interglacial supports fast future loss. Nature Climate Change, 10. 10.1038/s41558-020-0865-2
- Williams, Charles J.R., et al. (2020) CMIP6/PMIP4 simulations of the mid-Holocene and Last Interglacial using HadGEM3: comparison to the pre-industrial era, previous model versions, and proxy data. Climate of the Past, 16(22). 10.5194/cp-16-1429-2020

When constructing a paragraph about polar changes for the LIG, the Sime et al. (2025) reference summarizes much of what is required.
We deeply appreciate the reviewer for pointing out this. We will enrich our explanation of the current understanding of the importance of sea ice distribution in the MH and LIG in the revised manuscript, while referring to the previous studies mentioned by the reviewer.

L40: Change sentences to clarify that the focus is on ozone-climate feedbacks. Change to: "One possible factor that can affect the high latitudes is stratospheric ozone-climate feedbacks (Thompson and Wallace, 2000; Noda et al., 2017)."

We will revise the sentence in the revised manuscript as suggested by the reviewer.

L41: Rewrite to clarify the difference between human-generated ozone changes and the ozoneclimate feedbacks being investigated.

We will add a sentence to clarify the difference between human-generated ozone changes and the ozone-climate feedbacks in the past interglacials, as follows:

*"Although this mechanism was proposed for the human-induced ozone depletion at the poles, a similar ozone-climate mechanism may work in response to the different astronomical forcing in the past interglacials (Noda et al., 2017)."*

L47: Add numbers to show the size of the effects: ozone changes in ppm for present-day (human-induced) versus possible ppm changes for the MH or other past climates due to ozoneclimate feedbacks.

We will enrich our explanation about the present-day ozone change based on previous studies such as Son et al. (2010) while mentioning the magnitude of the ozone changes.

Son, S. W., Gerber, E. P., Perlwitz, J., Polvani, L. M., Gillett, N. P., Seo, K. H., ... & Yamashita, Y. (2010). Impact of stratospheric ozone on Southern Hemisphere circulation change: A multimodel assessment. Journal of Geophysical Research: Atmospheres, 115(D3).

L49: Again, ensure the previous estimated response size in ppm from Noda et al. is explicitly stated.

We will enrich our explanation about the magnitude of the changes in stratospheric ozone simulated by Noda et al. (2017) in the revised manuscript.

L53-66: Clearly spell out that the two objectives of this study are:
1. Testing whether a newer model yields the same results as the previous MH study.
2. Extending the work on interglacials from the MH to both the MH and LIG.

We will enrich our explanation of our objectives as pointed out by the reviewer, as follows:

*"In this study, we first test the response of the stratospheric ozone and its impact on climate during the MH using a state-of-the-art Earth system model. We then investigate the response of stratospheric ozone in both MH and LIG to show the impact of different astronomical forcing on the response of stratospheric ozone. For this purpose, we employed MRI-ESM2.0 and simulated…"*

Table 1 / Methods Section: The spin-up process is unclear. Is everything initiated from the same well-spun-up PI? Add comments on the usual spin-up duration (>250 years) and its importance for polar regions. Refer to Kageyama et al. (2021) for comments on this.

We appreciate the reviewer for pointing out this. The *PIcontrol* experiment was started the calculation from the well-spun-up PI condition submitted to CMIP6, while the *MHcontrol* and *LIGcontrol* experiments was started from the MH simulation submitted to CMIP6/PMIP4. We will clarify this while mentioning the potential effect of the length of the spin-up duration for polar regions in the revised manuscript.

L110-L115: Check whether this result is dependent on calendar adjustments and comment or adjust accordingly.

We expect that some of the three-months-mean-temperature would be strongly affected by the calendar adjustments. We will add a brief comment mentioning that this would be affected by the calendar adjustment in the revised manuscript. We also note here that we will move these mentions to Appendix, following the comments from another reviewer.

Figures 4, 7, and SIC-related figures: These figures should also show the actual PI and Interglacial SIE or SIC (add a 15% SIC line for each climate to each figure), not just anomalies. See Kageyama and Sime papers for why sea ice states/errors (in the PI and MH/LIG) are critical for determining SIC-climate changes (not just anomalies). Discuss any PI or MH/LIG sea ice state errors and their likely impacts.
We appreciate the reviewer for pointing out this. We will add a 15% SIC line in each figure. We also note here that the 15% SIC line for the PI condition was also shown in Yukimoto et al. (2019) (See their Figures 9 and 10). Yukimoto et al. stated as follows:

*"The Southern Hemisphere wintertime sea-ice extent (Fig. 9b) is slightly excessive in both models, particularly in MRI-ESM2.0. Both models simulate austral summer minimum sea-ice extents that coincide well with the observations" (From Yukimoto et al., 2019)*

and also as follows:

*"The sea-ice edge distribution of the Antarctic sea-ice distribution simulated by MRI-ESM2.0 (Figs. 10c, d) is reproduced fairly well." (From Yukimoto et al., 2019)*

As already mentioned in Yukimoto et al. (2019), the overall reproducibility of southern sea ice in the present-day condition by MRI-ESM2.0 is good. On this basis, we will further enrich our discussion about the impact of the sea ice distribution simulated by the model on the simulated MH and LIG climates. For example, in the LIG, one model (HadGEM3) reproduced the high surface temperature in the Arctic region that is consistent with reconstructions, which is associated with the loss of sea ice in summer (Guarino et al., 2020; Diamond et al., 2021), while other models did not simulate the loss of sea ice in the Arctic region in summer (Kageyama et al., 2021). Our model also does not simulate the loss of sea ice in the Arctic region, which may affect the strength of the ozone-climate feedback. For this reason, we will enrich our discussion about the sea ice distribution in PI, MH, and LIG simulated by our model, comparison with the reconstruction of the past temperature and sea ice distributions, and the potential impact of sea ice distributions to the ozone-climate feedback in both hemispheres in the revised manuscript.

Yukimoto, S., Kawai, H., Koshiro, T., Oshima, N., Yoshida, K., Urakawa, S., ... & Ishii, M. (2019). The Meteorological Research Institute Earth System Model version 2.0, MRI-ESM2. 0: Description and basic evaluation of the physical component. Journal of the Meteorological Society of Japan. Ser. II, 97(5), 931-965.

L133: Add more appropriate references and comments based on the MH/LIG sea ice and polar change papers listed above.
We will add references following the reviewer's comment in the above.

L133-151: This section is structured backward. It is primarily direct insolation impacts on Arctic sea ice that reduce the SIE (SIC), leading to warming and subsequent climate changes. See Diamond et al., Kageyama et al., and Sime et al. for clarification. Rewrite these paragraphs accordingly.
We will restructure the whole paragraph as suggested.

L168-171: Sentence is unclear—rewrite for clarity.
We will rewrite the sentence as follows:

*"The decrease in the ozone concentration would work in the direction to suppress the warming in the upper stratosphere, but the temperature increased as a result of the increase of shortwave radiation in austral winter during the MH and LIG."*

L171-175: Similar to the previous comment. The mixed tenses (previous interglacial times vs. previous Noda et al. results) make these lines difficult to parse. Separate:

- Climate-to-ozone feedback processes.
- Ozone-to-climate processes.
- MH/LIG simulation changes.
- Changes in the representation of climate-ozone-climate feedbacks.
- Differences between the Noda et al. results (previous model) and the new findings.

We agree with the reviewer. We will reorganize the sentence so the result of the present study and difference from the previous study become clear.

L240: Change "operate" to "occur."
Following the suggestion, we will change the wording here in the revised manuscript.

L243: "This contradicts the results shown by Noda et al. (2017), which suggest a warming in the Southern Hemisphere during the MH." This difference should be clearly stated in the abstract.
We will add a sentence explaining this in the Abstract, as follows:

*"These results are the opposite of the previous finding that implies the importance of ozone in southern hemisphere climates, indicating the need to determine whether this indicates model dependency."*

L245: Change "any season" to "all seasons" and "specifically" to "particularly."
We will rewrite the sentence following the suggestion.

L247: Contextualize the size of the changes relative to previously identified LIG and MH sea ice changes (Guarino, Sime, Chadwick, Gao, and Kageyama et al.).
Following the suggestion, we will enrich our discussion regarding the LIG and MH sea ice distributions and changes.

Figures 10 and 13: Show ozone-dependent impacts in K for pressures and latitudes (e.g. ~0.25K in the high Arctic). Explicitly state the magnitude of these numbers in the abstract and conclusions.
We will enrich our explanations about the magnitudes of the ozone change itself and its impact on air temperature in the revised manuscript.

L276: Spell out "mean annual" and "globally"—for example, large seasonal Arctic changes exist.
Following the suggestion, we will revise the phrase as *"...higher global and annual mean surface air temperature"*.

L281-L298: Since the study focuses on ozone impacts in polar/high-latitude regions, remove the discussion on global mean temperature and non-polar changes. Instead, discuss whether either model version accurately captures known MH/LIG sea ice and surface polar ocean changes and how that affects the climate-ozone-climate feedbacks.
We agree with the reviewer. We will decrease the amount of our explanation about the climate state simulated by MRI-ESM2.0 (following the comments by another reviewer) and remove the discussion regarding the global mean temperature and non-polar changes. Instead, we will enrich our discussion about the reproducibility of the reconstructions of temperature and sea-ice distribution around polar regions in our model in MH and LIG, and their potential impact on the climate-ozone feedback in the revised manuscript.

L299-L312: Clarify which aspects are model-specific (e.g., climate biases affecting interpretation) and state the headline results.

We will clarify that the response of ozone is robust between the previous study and our result, while the responses of wind patterns and surface air temperature can be dependent on models. We will enrich our explanation while mentioning the biases of the model, following the previous study (Yukimoto et al., 2019).

L313-L327: This paragraph is difficult to parse. If the argument is that further chemistry should be included in the model, first state which chemistry is currently missing, then explain why this could be important for MH, LIG, or another past climate interval.

We appreciate the reviewer for pointing out this. The intended arguments here were that the available reconstruction of the past atmospheric ozone is limited to the values near the surface and that the surface ozone concentration would be affected by processes that are not considered in the model, such as the wildfire-related supply of chemical components such as NOx and CH4. We will rewrite the paragraph in the revised manuscript.

---

## Author Response (AR1)

Summary: Watanabe et al. utilize MRI-ESM2.0, an Earth system model with an ozone module, to simulate climate and atmospheric ozone changes for the preindustrial (PI), mid-Holocene (MH), and Last Interglacial (LIG). Their study explores ozone-climate feedbacks by selectively enabling and disabling the ozone-chemistry module, with a particular focus on high-latitude regions. While their results indicate that stratospheric ozone changes can influence polar surface air temperatures, they suggest a limited impact on global mean temperature. The study extends previous work by examining both MH and LIG. It raises questions about model dependency and the need for further multi-model comparisons. The manuscript requires improvement in several areas, including the need for clearer differentiation between the sizes and sources of human-induced and natural ozone changes (i.e. quantification of ozone changes in ppm), and much more careful and robust discussion of sea ice state errors and their implications. Overall the study requires that it includes findings from previous research on MH/LIG polar climate changes, particularly regarding Arctic sea ice extent and its role in driving feedbacks. Further smaller clarifications are also required on spinup model-specific biases.

We appreciate the reviewer for the thoughtful review and for providing many valuable comments. In the course of revision, we have clarified that our model does not consider the human-induced effect on tropospheric ozone in numerical experiments conducted in this study. We have further clarified the distribution of the sea ice during the PI, MH, and LIG, and also compared our sea-ice distribution to the ones estimated by other climate models. We have also enriched our mention of the abundance and distribution of atmospheric ozone during the MH and LIG. We have taken all these comments and suggestions into account in the process of revision.

**Line-by-line comments:**

L9: Remove: "However, understanding the role of changes in stratospheric ozone during past warm interglacial periods is limited to MH conditions," since work suggests that previous understanding was incorrect

We agree with the reviewer. Instead of removing the sentence, we have rewritten the sentence as follows:

"However, little is known about the role of changes in the stratospheric ozone during past warm interglacial periods."

L13: Reverse clauses for better sentence construction: "We show that while ozone feedbacks may affect surface air temperature regionally, impacts on the zonal mean surface air temperature are small."

We have rewritten the sentence as follows:

"We further show that ozone feedbacks decrease the surface air temperature by ~0.35 and ~0.25 K in the high-latitude regions of the northern hemisphere in both MH and LIG, while the impact on the zonal mean surface air temperature around Antarctica was small."

L14: Add a sentence explaining that these results represent an update on previous findings or that further work using more models is needed to determine whether this indicates model dependency.

Following the suggestion, we have added the sentence as follows:

"This is the opposite of the previous finding that implies the importance of ozone in southern hemisphere climates, indicating the need to determine whether this indicates model dependency."

L14-15: Remove or rewrite the last sentence to reflect the previous comment.

Done.

**L16: Change "is expected" to "can."**

We have changed the expression as suggested.

L24-25: Split land-ocean versus sea ice-ocean feedbacks (Arctic versus other land) and rewrite the sentence clearly.

We have rewritten the sentence as follows:

"The warming is especially enhanced over high-latitude regions, possibly reflecting the amplification by feedback mechanisms such as sea-ice changes over the Arctic Ocean and vegetation feedback over land areas of the Northern Hemisphere ..."

L27: Remove the sentence on ice volume/GMSL changes. The authors do not address this here, and it adds nothing.

We agree with the reviewer. We have removed the sentence in the revised manuscript.

**L29-L34: These lines confuse the MH and LIG and add nothing. They can be removed.**

We have rewritten the sentence so that the discrepancy between temperature reconstruction and climate model exists in both MH and LIG, as follows:

"However, many climate models do not simulate the higher global annual mean surface air temperature inferred from the paleoclimate proxy during the MH and LIG (Masson-Delmotte et al. 2013; Otto-Bliesner et al., 2013; Liu et al. 2014; Brierley et al. 2020; Kaufman and Broadman 2023). The cause of this discrepancy has been vigorously debated (Liu et al. 2014, 2018; Hopcroft and Valdes 2019; Park et al. 2019; Bova et al. 2021; Zhang and Chen 2021; Thompson et al. 2022; Laepple et al. 2022; Kaufman and Broadman 2023), indicating that high-latitude processes over land and ocean are critical for further understanding the climate warming during the past interglacials."

L29-L34: Instead, provide a clear description of what is known about MH and LIG polar (sea ice and polar ocean) changes, particularly sea ice. Refer to:

- Gao, Qinggang, et al. (2025) Assessment of the southern polar and subpolar warming in the PMIP4 Last Interglacial simulations using paleoclimate data syntheses. Climate of the Past, 21. 10.5194/cp-21-419-2025
- Sime, Louise C., et al. (2025) More modest peak temperatures during the Last Interglacial for both Greenland and Antarctica suggested by multi-model isotope simulations. Climate of the Past [in review]. 10.5194/egusphere-2025-288
- Chadwick, Matthew, et al. (2023) Model-data comparison of Antarctic winter sea-ice extent and Southern Ocean sea-surface temperatures during Marine Isotope Stage 5e. Paleoceanography and Paleoclimatology, 38(11). 10.1029/2022PA004600
- Sime, Louise C., et al. (2023) Summer surface air temperature proxies point to near-seaice-free conditions in the Arctic at 127 ka. Climate of the Past, 19. 10.5194/cp-19-883-2023
- Diamond, Rachel, et al. (2021) The contribution of melt ponds to enhanced Arctic seaice melt during the Last Interglacial. The Cryosphere, 15(16). 10.5194/tc-15-5099-2021
- Kageyama, Masa, et al. (2021) A multi-model CMIP6-PMIP4 study of Arctic sea ice at 127 ka: Sea ice data compilation and model differences. Climate of the Past, 17(26). 10.5194/cp-17-37-2021
- Guarino, Maria Vittoria, et al. (2020) Sea-ice-free Arctic during the Last Interglacial supports fast future loss. Nature Climate Change, 10. 10.1038/s41558-020-0865-2
- Williams, Charles J.R., et al. (2020) CMIP6/PMIP4 simulations of the mid-Holocene and Last Interglacial using HadGEM3: comparison to the pre-industrial era, previous model versions, and proxy data. Climate of the Past, 16(22). 10.5194/cp-16-1429-2020

When constructing a paragraph about polar changes for the LIG, the Sime et al. (2025) reference summarizes much of what is required.

We deeply appreciate the reviewer for pointing this out. We have enriched our discussion and added references regarding the sea ice distribution during the MH and LIG in the revised manuscript, as follows:

"Indeed, the minimal extent of both Arctic and Antarctic regions would have been smaller than in the PI condition in the MH and LIG owing to the stronger insolation (Yoshimori and Suzuki, 2019; Brierley et al., 2020; Otto-Bliesner et al., 2020; Guarino et al., 2020; Diamond et al., 2021; Kageyama et al., 2021; Sime et al., 2023; Chadwick et al., 2023; Gao et al., 2025). Particularly, northern hemisphere sea ice may have completely melted seasonally during the LIG, which has a profound impact on the climate (Guarino et al., 2020; Diamond et al., 2021; Sime et al., 2023)."

L40: Change sentences to clarify that the focus is on ozone-climate feedbacks. Change to: "One possible factor that can affect the high latitudes is stratospheric ozone-climate feedbacks (Thompson and Wallace, 2000; Noda et al., 2017)."

We have rewritten the sentence in the revised manuscript as suggested by the reviewer, as follows:

"One possible factor that can affect the high latitudes is stratospheric ozone-climate feedbacks (Thompson and Wallace, 2000; Noda et al., 2017)."

L41: Rewrite to clarify the difference between human-generated ozone changes and the ozoneclimate feedbacks being investigated.

We have added a sentence to clarify the difference between human-generated ozone changes and the ozone-climate feedbacks in the past interglacials, as follows:

"Despite the uncertainty regarding the impact of the mechanism caused by the human-induced ozone depletion in the late 20th century, a similar ozone-climate mechanism may have worked in response to different astronomical forcing during past interglacials (Noda et al., 2017)."

Please see our reply to the next comment for further details.

L47: Add numbers to show the size of the effects: ozone changes in ppm for present-day (human-induced) versus possible ppm changes for the MH or other past climates due to ozoneclimate feedbacks.

After reassessing the impact of the mechanism in present-day conditions, we noticed that the magnitude of the ozone-climate relationship is uncertain even during the late 20th century (e.g., Langematz et al. 2018), although some idealized numerical simulations infer that this mechanism could work around Antarctica (e.g. Ferreira et al., 2015). For this reason, we have enriched our explanation to explain this, as follows:

"This has been pointed out to further affect the distribution of sea ice around Antarctica because the changes in jet strength affect the Ekman transport in the Southern Ocean (SO) and the warming of the surface ocean, while the plausibility and impact on the historical period remains still uncertain (Sigmond and Fyfe 2010, 2014; Bitz and Polvani 2012; Ferreira et al. 2015; Smith et al. 2012; Noda et al. 2017; Langematz et al. 2018). Despite the uncertainty regarding the impact of the mechanism caused by the human-induced ozone depletion in the late 20th century, a similar ozone-climate mechanism may have worked in response to different astronomical forcing during past interglacials (Noda et al., 2017)."

Instead, we have enriched our explanation about the ozone-climate impact of this mechanism during the MH (see our response to the next comment).

**L49: Again, ensure the previous estimated response size in ppm from Noda et al. is explicitly stated.**

We have enriched our explanation about the magnitude of the changes in stratospheric ozone simulated by Noda et al. (2017) in the revised manuscript, as follows.

"They showed that the positive ozone anomaly with a maximum value of ~0.2 ppm in the stratosphere would be caused by the different astronomical forcing during the MH. They further showed that this

would lead to the annual zonal mean surface air temperature anomaly of up to +1.7 K around the South Pole."

L53-66: Clearly spell out that the two objectives of this study are:

- 1. Testing whether a newer model yields the same results as the previous MH study.
- 2. Extending the work on interglacials from the MH to both the MH and LIG.

We have enriched our explanation of our objectives as pointed out by the reviewer, as follows:

"In this study, we first test the response of the stratospheric ozone and its impact on climate during the MH using a state-of-the-art Earth system model. We then investigate the response of stratospheric ozone in both MH and LIG to show the impact of different astronomical forcing."

Table 1 / Methods Section: The spin-up process is unclear. Is everything initiated from the same well-spun-up PI? Add comments on the usual spin-up duration (>250 years) and its importance for polar regions. Refer to Kageyama et al. (2021) for comments on this.

We appreciate the reviewer for pointing this out. The *Plcontrol* experiment was started with the calculation from the well-spun-up PI condition submitted to CMIP6, while the *MHcontrol* and *LIGcontrol* experiments were started from the MH simulation submitted to CMIP6/PMIP4. We have enriched our explanation as follows:

"We note that this spin-up period is relatively short compared to many of the PMIP4 experiments for these periods, and only two models have a spin-up period of 50 years (e.g., Kageyama et al., 2021). For this reason, the processes with long timescales, such as the oceanic circulation, would not have reached a steady state. Nevertheless, the response of total ozone during the MH and LIG had sufficiently reached a steady state after the spin-up period, so we consider this would not change our main conclusion."

**L110-L115: Check whether this result is dependent on calendar adjustments and comment or adjust accordingly.**

Yes, we expect that some of the three-months-mean-temperature can be strongly affected by the calendar adjustments. We clarified that all the monthly output was post-processed to apply calendar adjustments.

"The calendar adjustment was conducted in all the monthly output data for MH and LIG conditions outputs to consider the changes in the length of months (Bartlein and Shafer 2018)."

Figures 4, 7, and SIC-related figures: These figures should also show the actual PI and Interglacial SIE or SIC (add a 15% SIC line for each climate to each figure), not just anomalies. See Kageyama and Sime papers for why sea ice states/errors (in the PI and MH/LIG) are critical for determining SIC-climate changes (not just anomalies). Discuss any PI or MH/LIG sea ice state errors and their likely impacts.

We appreciate the reviewer for bringing this to our attention. We have added a 15% SIC line in each figure. As previously discussed in Yukimoto et al. (2019), the overall reproducibility of southern sea ice in the present-day condition by MRI-ESM2.0 is good. On the other hand, in the LIG condition, one model (HadGEM3) reproduced the high surface temperature in the Arctic region that is consistent with reconstructions, which is associated with the loss of sea ice in summer (Guarino et al., 2020; Diamond et al., 2021), while other models did not simulate the loss of sea ice in the Arctic region in summer (Kageyama et al., 2021). This may also affect the strength of the ozone-climate interaction. During this revision, we have further enriched our discussion about the impact of the sea ice distribution simulated by the model on the ozone-climate interactions in Discussion.

"This different response around Antarctica may be related to the different biases in temperature. For example, the surface air temperature around Antarctica simulated by MRI-ESM2.0 employed in this study exhibits a cool bias, while that simulated by MRI-CGCM3, which is an atmospheric general circulation model used in MRI-ESM1 employed by Noda et al. (2017), exhibits a warm bias, especially over the Southern Ocean (Yukimoto et al., 2019). As a result, the responses of the sea ice distribution and the surface air temperature differed from those reported by Noda et al. (2017), exhibiting zonally asymmetric characteristics. In MRI-ESM2.0, the sea-ice extent around Antarctica is slightly higher than observations and MRI-CGCM3, but it reproduces the sea-ice edge around Antarctica well (Yukimoto et al., 2019). The reproduction of sea ice extent in the Arctic Ocean in MRI-ESM2.0 was much closer to the observations than that of MRI-CGCM3. These different sea-ice distributions may have also contributed to the different responses between the models. Specifically, for the case of the LIG, the sea ice may have completely melted seasonally (Guarino et al., 2020; Diamond et al., 2021; Sime et al., 2023). In this case, the ozone-induced climatic impact of the changes in sea-ice distributions in the Arctic Ocean may have been smaller than in our model."

L133: Add more appropriate references and comments based on the MH/LIG sea ice and polar change papers listed above.

Please see our reply to the next comment.

L133-151: This section is structured backward. It is primarily direct insolation impacts on Arctic sea ice that reduce the SIE (SIC), leading to warming and subsequent climate changes. See Diamond et al., Kageyama et al., and Sime et al. for clarification. Rewrite these paragraphs accordingly.

We have reorganized the paragraph as follows, while mentioning the previous study discussing the LIG sea ice extent. We note here that this part is moved to Appendix A in the revised manuscript.

"For the LIG case, the overall patterns of the surface air temperature anomaly from the PI conditions were similar to the MH, but the signal tended to be stronger, which was similar to the previous study (Otto-Bliesner et al. 2021). The sea ice concentration during SON decreased over a wide area of the Arctic Ocean (Fig. A3g). The ice edge retreated compared with the PI in the Greenland Sea and Barents Sea. However, it did not exhibit the complete loss of sea ice simulated by previous studies (Guarino et al., 2020; Diamond et al., 2021; Sime et al., 2023) as in many other climate models (Kageyama et al., 2021). Nevertheless, significant warming was observed in JJA and SON (right panels in Fig. 2a), reflecting the higher insolation anomaly and the associated response of sea ice distribution (Figs. 3b and A3). In SON, specifically, the surface air temperature over the Arctic Ocean increases by more than ~4 K compared with the PI conditions (Fig. A4d). As a result, significant warming is observed in the annual mean temperature over the regions covering the Greenland Sea, Norwegian Sea, and Barents Sea (Fig. A1c). The warming is also observed in the central part of North America, the southern part of South America and Africa, the northern part of Australia, the western part of the Eurasian continent, and in part of the Southern Ocean. Despite the warming in these regions, the global annual mean surface air temperature did not exceed the PI value."

**L168-171: Sentence is unclear—rewrite for clarity.**

We have rewritten the sentence as follows:

"The decrease in the ozone concentration would work in the direction of suppressing the warming in the upper stratosphere, but the temperature increased because of the increase in shortwave radiation during austral winter in the MH and LIG."

L171-175: Similar to the previous comment. The mixed tenses (previous interglacial times vs. previous Noda et al. results) make these lines difficult to parse. Separate:

- Climate-to-ozone feedback processes.
- Ozone-to-climate processes.
- MH/LIG simulation changes.
- Changes in the representation of climate-ozone-climate feedbacks.
- Differences between the Noda et al. results (previous model) and the new findings.

We have reorganized the paragraph so that the results of the present study and differences from Noda et al. (2017) become clear. Please see Sections 3.1 and 3.2 in the revised manuscript.

**L240: Change "operate" to "occur."**

Following the suggestion, we have revised the wording in this section of the manuscript.

L243: "This contradicts the results shown by Noda et al. (2017), which suggest a warming in the Southern Hemisphere during the MH." This difference should be clearly stated in the abstract.

We have added a sentence explaining this in the Abstract, as follows:

"This is the opposite of the previous finding that implies the importance of ozone in southern hemisphere climates, indicating the need to determine whether this indicates model dependency."

L245: Change "any season" to "all seasons" and "specifically" to "particularly."

We have revised the wording according to the suggestion.

L247: Contextualize the size of the changes relative to previously identified LIG and MH sea ice changes (Guarino, Sime, Chadwick, Gao, and Kageyama et al.).

Following the suggestion, we have enriched our discussion regarding the LIG and MH sea ice distributions and changes, as follows:

"In MRI-ESM2.0, the sea-ice extent around Antarctica is slightly higher than observations and MRI-CGCM3, but it reproduces the sea-ice edge around Antarctica well (Yukimoto et al., 2019). The reproduction of sea ice extent in the Arctic Ocean in MRI-ESM2.0 was much closer to the observations than that of MRI-CGCM3. These different sea-ice distributions may have also contributed to the different responses between the models. Specifically, for the case of the LIG, the sea ice may have completely melted seasonally (Guarino et al., 2020; Diamond et al., 2021; Sime et al., 2023). In this case, the ozone-induced climatic impact of the changes in sea-ice distributions in the Arctic Ocean may have been smaller than in our model."

Figures 10 and 13: Show ozone-dependent impacts in K for pressures and latitudes (e.g. ~0.25K in the high Arctic). Explicitly state the magnitude of these numbers in the abstract and conclusions.

We have added a new figure to show the ozone-dependent impacts in K for pressures and latitudes in Figure 4. We have mentioned these numbers in the abstract and conclusions.

L276: Spell out "mean annual" and "globally"—for example, large seasonal Arctic changes exist.

Following the suggestion, we have revised the phrase as "...higher global and annual mean surface air temperature".

L281-L298: Since the study focuses on ozone impacts in polar/high-latitude regions, remove the discussion on global mean temperature and non-polar changes. Instead, discuss whether either model version accurately captures known MH/LIG sea ice and surface polar ocean changes and how that affects the climate-ozone-climate feedbacks.

Following the suggestion, we have moved the discussion about the global climate states in MH and LIG to Appendix A. Instead, we have enriched our discussion about the different responses of the

ozone-induced impact between this study and the previous study (Noda et al., 2017) and the effect of the sea-ice distributions in the revised manuscript, as follows:

"This different response around Antarctica may be related to the different biases in temperature. For example, the surface air temperature around Antarctica simulated by MRI-ESM2.0 employed in this study exhibits a cool bias, while that simulated by MRI-CGCM3, which is an atmospheric general circulation model used in MRI-ESM1 employed by Noda et al. (2017), exhibits a warm bias, especially over the Southern Ocean (Yukimoto et al., 2019). As a result, the responses of the sea ice distribution and the surface air temperature differed from those reported by Noda et al. (2017), exhibiting zonally asymmetric characteristics. In MRI-ESM2.0, the sea-ice extent around Antarctica is slightly higher than observations and MRI-CGCM3, but it reproduces the sea-ice edge around Antarctica well (Yukimoto et al., 2019). The reproduction of sea ice extent in the Arctic Ocean in MRI-ESM2.0 was much closer to the observations than that of MRI-CGCM3. These different sea-ice distributions may have also contributed to the different responses between the models. Specifically, for the case of the LIG, the sea ice may have completely melted seasonally (Guarino et al., 2020; Diamond et al., 2021; Sime et al., 2023). In this case, the ozone-induced climatic impact of the changes in sea-ice distributions in the Arctic Ocean may have been smaller than in our model."

L299-L312: Clarify which aspects are model-specific (e.g., climate biases affecting interpretation) and state the headline results.

We have enriched our discussion so that the potential impact of the model biases is mentioned, following the previous study (Yukimoto et al., 2019).

"This different response around Antarctica may be related to the different biases in temperature. For example, the surface air temperature around Antarctica simulated by MRI-ESM2.0 employed in this study exhibits a cool bias, while that simulated by MRI-CGCM3, which is an atmospheric general circulation model used in MRI-ESM1 employed by Noda et al. (2017), exhibits a warm bias, especially over the Southern Ocean (Yukimoto et al., 2019)."

L313-L327: This paragraph is difficult to parse. If the argument is that further chemistry should be included in the model, first state which chemistry is currently missing, then explain why this could be important for MH, LIG, or another past climate interval.

The intended arguments here were that the available reconstruction of the past atmospheric ozone is limited to the values near the surface and that the surface ozone concentration would be affected by processes that are not considered in the model, such as the wildfire-related supply of chemical components such as NOx and CH4. Because the original paragraph was misleading, we have rewritten the paragraph in the revised manuscript, as follows:

"While the reconstruction of stratospheric ozone in the past is not available, the tropospheric ozone burden during the LIG was estimated based on a record of the clumped isotope composition of O2 in the East Antarctic ice core (Yan et al. 2022), indicating a reduction in the tropospheric ozone burden by nearly 9 % compared with the PI conditions. It has been inferred that the dispersal of modern humans had not yet occurred during the LIG (Liu et al. 2015; Malaspinas et al. 2016; Groucutt et al. 2018), which makes the LIG an ideal period for reconstructing the tropospheric ozone under a smaller impact from the influence of humans (Yan et al. 2022). However, the tropospheric ozone concentration may be affected not only by factors such as astronomical forcing, but the activity of early humans. For example, early humans could affect the supply rate of methane to the troposphere through rice cultivation. Indeed, ice core records indicate that the atmospheric methane level has increased since the MH compared with the LIG (Spahni et al. 2005; Singarayer et al. 2011), which would be associated with rice cultivation (Ruddiman 2003; Ruddiman et al. 2008) and/or natural wetland emissions (Schmidt and Shindell 2004; Sowers 2010; Singarayer et al. 2011). The early

human activity can affect the supply rate of NOx to the troposphere by causing wildfire events (Ward et al. 2012). The magnitude of wildfire events is determined by the complex interrelationship between climate, vegetation activities, and early human activities, so it would also be affected by the different astronomical forcing. Because our model does not consider the different source fluxes of these gases, the simulated tropospheric ozone should not be compared with the reconstructions. Simulations of the last glacial cycle using a fully coupled Earth system model including the atmosphere, ocean, aerosol, ozone chemistry, and vegetation cycle would be ideal for understanding the variations in the tropospheric ozone distribution during this period."

**Anonymous Referee #2**

Review. This paper explores the effect of stratospheric ozone changes – if any – on the climate of the mid-Holocene (MH) and the Last Interglacial (LIG). The question is well posed, and most interesting. To answer it the authors have performed an excellent set of model runs: 3 epochs (1850 PI control, MH and LIG) and, for each epoch, 2 runs (with PI ozone and with interactive ozone). So, they should be able to answer the question clearly.

We appreciate the reviewer for the thoughtful review and for suggesting the improvement of the structure of the manuscript. We reorganized the manuscript by moving some of the figures describing the climatic condition simulated by the model to the Appendix, so that our main results and discussion, that is, atmospheric ozone distributions during the MH and LIG, appear earlier in the manuscript. We enriched our discussion regarding the atmospheric ozone during the MH and LIG by adding some figures, such as a latitude-altitude plot of atmospheric ozone, which is crucial for understanding the dynamics of ozone. We have taken all these comments and suggestions into account in the process of revision.

Unfortunately the manuscript, in its current form, is really a mess. The authors waste 7 figures (each with many panels) discussing all manner of secondary considerations, and only show an ozone field for the first time in Figure 8. So, the narrative is completely backwards. If one is trying to tell the impact of ozone changes, one should start by showing the ozone changes. But this need to be done properly. Why are we shown ozone at the 3 hPa level in Figure 8? Is that were the ozone layer is? I would imagine the readers want to see ozone at 50 or 70 hPa. What about a latitude/height map of ozone changes in the ML and LIG? Or again, how about showing a lat/lon map of total column ozone (TCO)? Is the ozone layer thicker or thinner than under PI forcings? By how many Dobson units? I have no idea what the answer is (as I have not run the models), but none of this is shown in the paper. Again: the paper needs to start with 2 or 3 well chosen figures telling us what ozone looks like in the MH and LIG, and how it differs from the PI control.

We agree with the reviewer's suggestion. We have largely reorganized the manuscript in this revision. Now all the climate-related figures are moved to Appendix A, and the results start with explaining the total mass of ozone in these periods, as follows:

"The latitudinal-vertical plot of the zonal-mean ozone concentration anomaly for the MH and LIG conditions from the PI condition is shown in Fig. 2a. For the case of the MH, the ozone concentration was higher compared to PI in the lower stratosphere at low-latitude regions and near the South Pole, while it was lower in the upper stratosphere at low-latitude regions and near the North Pole. As a result, the 150-year mean value of total ozone was 314.7 DU, which was slightly lower than the PI value (315.4 DU). For the case of the LIG, on the other hand, the 150-year mean value of total ozone was 316.7 DU, which was slightly higher than in the PI. This is related to the positive anomaly in ozone concentration in broad regions of the lower stratosphere (Fig. 2b). These anomalies would be primarily associated with the different astronomical forcing during the MH and LIG."

We have also added a new latitude-height map of the ozone anomaly in the MH and LIG (in the revised Figure 2). We have also added the ozone concentration in the MH and LIG using contours in this figure,

showing that the ozone layer is present in both MH and LIG. We have also added a new figure showing the latitude-height map of the air temperature anomaly and ozone-induced air temperature anomaly in the new Figure 4. We have also added a vertical-seasonal plot of ozone and temperature in tropical regions to discuss the response of the ozone distribution according to the different astronomical forcing in the MH and LIG, and the dependencies of ozone response in the tropics and the high-latitude regions of the northern and southern hemispheres (Figure 5). We believe these additions would help the readers to understand the basic responses of the ozone distributions and their potential impact on climate during the MH and LIG.

Next, the key results are at the very end of the paper, in Figures 12, 13 and 14: these show that ozone changes in the ML and LIG have basically no statistically significant impact on surface climate. So, why are the most important figures left at the end of the paper? And why are the authors not stating clearly that the effect of interactive on surface temperature are minuscule? And what about precipitation (which is not show)? I suspect ozone is also irrelevant for the that. In my mind that should be the key point of the paper: ozone changes in the ML and LIG don't matter. It is a null result, but null results are very much worth publishing. In all honesty, I am not surprised that ozone changes make no difference: this is because I suspect these changes are small. It takes something like an ozone hole over the South Pole (as we have seen in the late 20th century) to make a substantial climate impact. Hence the key figure the readers need to see: how big are the ozone changes in the ML and LIG compared to those caused by CFCs?

As in our reply to the previous comment, we have reorganized the manuscript. As a result, the original Figure 13 appears much earlier in the manuscript. After showing that the ozone-induced climate change is minor (we note, however, it affects in north of 60°N), we explained the details of the impact in the regions around the North and South poles. We have also clarified this in the abstract and conclusions, as follows:

"We further show that ozone feedbacks decrease the surface air temperature by ~0.35 and ~0.25 K in the high-latitude regions of the northern hemisphere in both MH and LIG, while the impact on the zonal mean surface air temperature around Antarctica was small. This is the opposite of the previous finding that implies the importance of ozone in southern hemisphere climates, indicating the need to determine whether this indicates model dependency."

The ozone-induced response of precipitation around Antarctica was also minor. The magnitude of the change in the total ozone in the atmosphere from PI was 2 DU even in the LIG, which is much smaller than the changes in the late 20th century. This also implies that the ozone changes to astronomical forcing are minor, as the reviewer notes here. However, another important point is that the astronomical forcing affects the ozone concentration seasonally, so the impact of ozone on the seasonal changes of the atmospheric structure is also important. For this reason, we have enriched our discussion regarding the vertical changes in ozone and temperature anomalies in different regions (the tropics and the high-latitude regions of the northern and southern hemispheres) by adding figures showing vertical-seasonal plots (Figs. 5, 7, and 12).

Recommendation. The paper – in its present form – should be rejected. However, the authors should be strongly encouraged to resubmit. They have a nice set of runs, and a very clean story to tell: the ozone changes in the ML and LIG are small, and therefore they make little difference for the surface climate. Such a paper is easy to write, as the key points can be made with a few simple figures, and no complicated mechanisms needed to be invoked. It will be a nice contribution to the literature. I look forward to it.

We appreciate the reviewer for the thoughtful comment. Yes, as mentioned above, we reorganized the structure of the manuscript so that the ozone distributions during the MH and LIG and their impact on climatic conditions are clearly shown in the manuscript.

**Community comments by Jo-Jo Eumerus**

**A few minor observations:**

We appreciate Jo-Jo Eumerus for providing valuable comments on our manuscript.

\* Are there any data that identify causal mechanisms e.g are the ozone and temperature changes solely due to local insolation and sea ice or are there atmospheric circulation changes (perhaps sector-dependent) that also play a role?

We expect that the signal of local temperature anomaly caused by ozone over the southern polar region would have been caused by a complex interplay of ozone chemistry, atmospheric circulation, radiation, and/or cloud formation instead of a simple response to local insolation. For this reason, the exact mechanism would not be easy to identify clearly in this case, partly because the impact on climate would be very small (i.e., significant signal is limited seasonally and regionally in the southern hemisphere). This is in contrast to the case of the previous study (Noda et al., 2017), which showed an ozone-climate feedback caused by the wind and sea ice responses in the MH. This different response may be attributed to the biases of the model we used and the one used in Noda et al. (2017), so we have enriched our discussion regarding this in the revised manuscript in Discussion. For further quantitative constraints on the mechanism driving the climatic impact of atmospheric ozone, investigations using multiple Earth system models with interactive ozone would be important.

**\* In Fig3, it seems like MH and LIG have opposite temperature changes in Antarctica during JJA; any indication why?**

We expect that the different temperature responses around Antarctica between MH and LIG would be associated with the different astronomical forcing (Figure 5). The seasonal timing of the onset of the positive insolation anomaly in the regions around the Southern Ocean is March-April in the LIG, which is much earlier than in the MH.

**\* Has ozone effects on climate just in Antarctica?**

The changes in ozone may affect the temperature in the northern hemisphere in the MH and LIG, as can be seen in the difference between grey and red/blue lines in Figure 6. In this revision, we have enriched our discussion regarding the ozone impact on the northern hemisphere (Section 3.3).

---

## Author Response (AR2)

The paper by Watanabe and co-authors present simulations of the effects of Earth's orbital variations on climate with a focus on how ozone may affect the ability of models to reproduce the reconstructed climate of the mid-Holocene (MH) and Last Interglacial (LIG) warm periods. The analysis builds on, and in some aspects conflicts with, previous work (Noda et al., 2017) that used an earlier version of the MRI model but only looked at the MH. The authors find some significant and physically consistent changes in ozone, temperature and winds in the stratosphere but the impact in the troposphere and, in particular, on the surface climate are relatively modest.

I was not one of the reviewers for the first submitted version of the paper, nor have I read that version. I have, however, read the comments from the reviewers (after having read the current version) and feel that the revisions to the structure and presentation of the results may have gone some way to address the concerns from reviewer 2 of the first version. I certainly did not have similar concerns about the presentation when reading the paper.

We are grateful to the reviewer for the comments and suggestions, which have helped us to improve the manuscript. As indicated in the responses that follow, we have taken all these comments and suggestions into account during the revision process.

My one significant concern about the discussion of the results centers around the near-surface wind response around the South Pole shown in Figure 9 and discussed starting around line 239. Particularly striking is the seasonal change in the response, with strong positive anomalies for the LIG simulation in JJA-SON, and strong and relatively symmetric weakening in DJF and MAM. I can see the physical explanation for the acceleration of the near-surface winds for JJA and SON being connected to similar signed changes in the stratosphere in a way that is quite analogous with what is seen for ozone depletion since the 1980s. But the changes in DJF and MAM are much harder to connect with the temperature changes that have been presented. The seasonal cycle of temperature change at 3hPa (Figure 3) does not show much of a change in temperature gradient across the mid-to-high latitudes of the Southern Hemisphere for DJF and MAM so I suspect there is some change in temperature that is not apparent in the figures that is driving the weakening of the winds. I would like to see just a little bit more supporting information or discussion to provide an explanation for the strong seasonal behaviour of the impact on the near-surface winds around the Antarctic.

We appreciate the reviewer for pointing this out. As mentioned by the reviewer, the acceleration of the near-surface wind patterns for JJA and SON is associated with the temperature changes in the stratosphere, which is originally driven by the different astronomical forcing. On the other hand, the weakening of the near-surface wind patterns for DJF and MAM originates from the upper troposphere, reflecting the cooling of the troposphere in the tropics (Fig. 5a and 5b). Because this cooling is not as strong as in the mid-high latitude regions in the southern hemisphere (Fig. 7a and 7b), this decreases the meridional temperature gradient in the troposphere, weakening the surface wind in DJF and MAM around Antarctica. We have enriched our discussion as follows:

"This wind response would be caused primarily by the decrease of the meridional temperature gradient in the troposphere because of the cooling of the troposphere in the tropics in DJF and MAM (Fig. 5a and 5b), which is stronger than that in the high-latitude region (Fig. 7a and 7b)."

My other comments are minor and are itemized below.

Line 57 – this study extends previous study of Noda et al., (2017) that used MRI-ESM1 to investigate ozone effects during the MH.

Yes, our study extends the result of Noda et al. (2017) to not only the MH condition but also the LIG condition using the MRI-ESM2.0.

Lines 57 – 62: The section beginning 'This study showed that the positive ozone...' through to '...different astronomical forcing during the MH.' has a few instances of repetition that could be removed to make it a bit easier reading.

We have rewritten the sentence as follows:

"This study showed that the positive ozone anomaly in the upper stratosphere around the South Pole during the austral summer of the MH would propagate to the lower stratosphere in the austral winter. This anomaly would increase the air temperature and weaken the southern westerly jet. They further suggested that it may contribute to the retreat of sea ice in the Southern Ocean during the MH. They showed that the positive ozone anomaly with a maximum value of ~0.2 ppm in the stratosphere would be caused by the different astronomical forcing during the MH."

Lines 89 - 90: The MH and LIG simulations started 'from the steady state obtained for the MH condition submitted to PMIP4.' It is not quite clear what the 'steady state' from the PMIP4 simulation is. I assume the simulations presented here start from the model state after the first 51 years of spin-up for the PMIP4 simulation, but as the authors have explained this is not necessarily a steady state.

The reviewer is correct for pointing this out. We have reworded the sentence as follows:

"...starting from the calculation of the MH condition submitted to PMIP4"

Line 101 – Table 1: In the greenhouse gas column there is reference to NO2, but I am certain this is a typo and should be N2O.

Corrected.

Lines 118 – 126: The discussion of the temperature effects on ozone needs to be strengthened. I am most familiar with the temperature effects due to changes in CO2, for which the impacts on ozone through the temperature sensitivity of ozone loss reactions is well known. Here the irradiance is changing so in addition to the temperature effects on O3 + O -> 2O2, there would be changes on O2 and O3 photolysis. Given the strong anti-correlation of temperature and ozone changes, I am not doubting that the temperature effect dominates only that the discussion is a bit underdeveloped.

We agree with the reviewer. We have enriched our discussion in the revised manuscript, as follows:

"Although the insolation anomaly in the MH and LIG would also affect the photochemical reaction rates of Reactions R2 and R4, the negative correlation indicates that the temperature effect of Reaction R1 would dominate the behavior of stratospheric ozone anomalies in the MH and LIG."

Line 152 – Figure 4: The presentation of annual mean zonal cross section changes really dilutes the impacts of the changes. The seasonal cycle of changes at 3hPa shown in Figure 3 has the LIG simulation with much larger changes than those seen for the MH, but in the annual average these appear weaker. In particular, the strong positive changes in temperature seen near the north pole in JJA for Figure 3d are now negative in the annual

average shown in Figure 4b. Is the manuscript losing important aspects of the response by presenting annual average changes? Because of the nature of the changes in forcings I can appreciate that it is not as simple as showing JJA and DJF – the insolation anomaly in the MH maximizes in JJA over the northern high latitudes but in SON for high southern latitudes.

The reviewer is correct in pointing out that seasonal changes are important for ozone dynamics in the MH and LIG. This figure was added to illustrate that the comprehensive view of ozone distribution (e.g., the presence of the ozone layer in the stratosphere) remains largely unchanged, even in the MH and LIG, following the previous review comment. Because we have already been extensively discussed in the manuscript, we have retained Figure 4 in the revised manuscript.

Line 170: The use of the word 'transported' in 'This signal was transported into the lower stratosphere.' seems to suggest there was advection of the warm anomaly seen earlier in the year at lower pressure downwards. I think this is difficult to justify. The warm anomaly does seem to follow the ozone anomaly, so I would think it is more of a perturbation of local heating rates.

Because the word "transported" was misleading, we replaced it with the word "propagated", as follows:

"This signal propagates into the lower stratosphere. For the case of the LIG, the positive ozone anomaly originated from the upper stratosphere between January and May is present in the lower stratosphere between February and August, whose impact is larger than 1 K. This signal may partly propagate into the troposphere, ..."

Lines 176– 183: The authors might make it clearer that the discussion is focusing on the differences between the differences in the full simulations and the nochem simulations.

We have enriched our explanation in the revised manuscript, as follows:

"Comparing the air temperature anomalies obtained for the control experiments and those obtained for the noChem experiments, the impact of the atmospheric ozone changes on the zonally averaged climate was small in both the MH and LIG in the south of 60°N."

Lines 233 - 236: I believe I understand the explanation the authors are suggesting for the acceleration of the southern westerly jet. If I do, the authors could make reference to Figure 3c and 3d to show the stronger meridional temperature gradient driven by warming on the equator-ward side of the jet that is responsible for the acceleration.

We have enriched our explanation in the revised manuscript, as follows:

"The westerly jet intensity is related to the meridional air temperature gradient in the stratosphere at the mid-latitude regions, which is attributed to the larger meridional temperature anomaly owing to the different insolation forcing (Fig. 3c and 3d)."

Lines 299 – 300: Similar to the comment on line 170 about the use of 'transported' when discussing temperature anomalies, is it the temperature transported or the ozone anomaly that carries along with it a heating rate anomaly?

Because the word "transported" was misleading, we replaced it with the word "propagated".

Line 301: The statement 'possibly reflecting the modulation of the Arctic Oscillation.' seems unsupported by analysis. And I am not sure what the authors wish to imply with this statement.

The original sentence was misleading, so this sentence was removed in the course of this revision.

Line 303: 'This ozone-induced expansion of sea ice...' The caption for Figure 13 states that the dashed line is the sea-ice edge for the Plcontrol, while the solid line is for the MHcontrol or LIGcontrol experiment. When you look closely at Figure 13g, the dashed and solid line seems to suggest that the sea-ice has retreated in the Atlantic side of the Arctic between Plcontrol and LIGcontrol, but the ozone related temperature changes are negative. The idea that the ozone-driven cooling is partly counteracting an insolation-driven warming is supported by Figure 12. But then can it be deduced that the ozone-driven changes are also responsible for counteracting some of the sea-ice retreat?

We appreciate the reviewer for pointing this out. The sea ice edge in the LIG, for example, is smaller than that in the PI in SON, as shown in the figure, reflecting the stronger insolation forcing in summer in the LIG. On the other hand, the impact of changes in ozone distributions works to suppress the insolation-driven reduction in sea ice distribution. Because the original sentence was misleading, we have rewritten the sentence as follows:

"While the Arctic sea-ice distributions in the MH and LIG are smaller than in the PI in SON owing to the effect of the different astronomical forcing (black solid lines in Fig.13), the cooling effect from the changes in ozone distributions works to suppress the reduction in sea-ice distribution due to the different astronomical forcing. This ozone-induced net expansion of sea ice would be..."

Line 311: Figure 12 is for the North pole but the caption has 'at the tropic region (60–90°N)'. Corrected.

**Comments from Editor:**

Abstract:

L. 15: or "MH and LIG, respectively"?

Corrected.

L. 16: "is" small

Corrected.

Last sentence of abstract needs to be rephrased by making it more specific/less vague.

We have rewritten the sentence as follows:

"This is the opposite of the previous finding that implies the importance of ozone in southern hemisphere climates, indicating the need for further assessment of how dynamic ozone variations affect climate and atmospheric structures during past warm interglacial periods using multiple Earth system models."

L. 35-36:something is missing here. This sentence does not make sense. Maybe you are referring to sea-ice extent?

We appreciate the editor for finding this mistake. We have rewritten the sentence as follows: "Indeed, the minimal extent of sea ice in both Arctic and Antarctic regions would have been smaller than in the PI condition in the MH and LIG owing to the stronger insolation"

L. 242: Results section should be in present tense.

Corrected throughout the Results section.

L. 243-245 and 249-250: Are you describing your results or those of Noda et al? You might want to move all the comparisons to Noda et al., to the discussion.

Following the suggestion, we have moved these sentences into the Discussion section.

Caption of figure 12: 60N-90N are not the tropics.

Corrected.

Conclusions: I suggest to choose a tense (past or present) and use it throughout the Conclusion.

We have chosen the present tense in the Conclusions.

L. 374: Add "during the MH and LIG".

Done.